# Moisture-tolerant Mg-metal electrodes for practical fabrication of rechargeable Mg batteries

Woo Joo No[1,2,11], Jonghyun Han [1,11], Jinyeon Hwang[1,11], Sibylle Riedel[3], Minji Jeong[1], Hyeong Kyu Park[4], Ju Young Kim[2,5], Kwan Young Lee [2], Minah Lee [1], Taeeun Yim [6], Hyun Deog Yoo [7], Hyung Chul Ham [4], Sang-Young Lee [8], Zhirong Zhao-Karger [3,9] & Si Hyoung Oh [1,10] ✉

Despite striking advantages in terms of cost and safety, penetration of rechargeable Mg batteries into the commercial market is still hampered by major technical challenges, including intrinsic hypersensitivity of Mg metal to moisture, which readily forms a compact ion-insulating film on the surface. To unlock this critical constraint, a moisture-tolerant Mg electrode is developed that is capable of efficient Mg plating-stripping even in the highly moist electrolytes. Here we show that short immersion of Mg metal in trimethyl phosphate creates a sacrificial protection layer containing dimethyl magnesium in magnesium dimethyl phosphate, which synergistically scavenges water molecules from the electrolytes instantly, enabling manufacturing of Mg-ion cells under moist and/or atmospheric conditions. This simple and scalable strategy provides a practical route to reducing the manufacturing costs of rechargeable Mg batteries, thereby expediting their early commercialization.

As the carbon dioxide level in the atmosphere has recently exceeded 400 ppm for the first time since the Pliocene Era, global climate change has become a matter of human survival[1]. This has driven the global energy sector to expand green renewable energy sources, where better energy storage systems (ESSs) with higher cost competitiveness, environmental sustainability, and safety than conventional Li-ion batteries (LIBs) are urgently required to expedite its growth. Owing to their natural abundance, high capacity, and intrinsic chemical stability, rechargeable batteries based on magnesium resources (RMBs) have been extensively studied as a serious alternative to LIBs that can potentially meet the strict requirements for ESS markets[2–4]. Those

tireless efforts for the last three decades have led to the recent surge of remarkable scientific advances in the electrode materials[5–15] and the electrolytes[16–31].

In terms of commercial viability, however, this immense opportunity for the emerging battery market was hampered by several technical challenges in electrode materials and electrolytes, including the extreme chemical vulnerability of Mg metals to moisture, where even a single exposure to a trace amount of water contaminant during the battery manufacturing process could be fatal to battery performance[32,33]. This is attributed to the spontaneous formation of a compact, ion-blocking passive layer on the surface of Mg metals

[1]Energy Storage Research Center, Korea Institute of Science Technology (KIST), Hwarang-ro 14-gil 5, Seongbuk-gu, Seoul, Republic of Korea. [2]Department of Chemical and Biological Engineering, Korea University, 145, Anam-ro, Seongbuk-gu, Seoul, Republic of Korea. [3]Helmholtz Institute Ulm (HIU), Electrochemical Storage, Helmholtzstrasse 11, Ulm, Germany. [4]Department of Chemical Engineering, Inha University, Inha-ro 100, Michuhol-gu, Incheon, Republic of Korea. [5]Clean Energy Research Center, Korea Institute of Science Technology, Hwarang-ro 14-gil 5, Seongbuk-gu, Seoul, Republic of Korea. [6]Department of Chemistry, Incheon National University, 119 Academy-ro, Yeonsu-gu, Incheon, Republic of Korea. [7]Department of Chemistry and Chemical Institute for Functional Materials, Pusan National University, Busan, Republic of Korea. [8]Department of Chemical and Biomolecular Engineering, Yonsei University, 262 Seongsanno, Seodaemun-gu, Seoul, Korea. [9]Institute of Nanotechnology (INT), Karlsruhe Institute of Technology (KIT), Hermann-von-Helmholtz Platz 1, Eggenstein-Leopoldshafen, Germany. [10]Division of Energy & Environment Technology, Korea University of Science and Technology, Hwarang-ro 14-gil 5, Seongbuk-gu, Seoul, Republic of Korea. [11]These authors contributed equally: Woo Joo No, Jonghyun Han, Jinyeon Hwang. ✉e-mail: sho74@kist.re.kr

immediately upon exposure to moist electrolytes or humid air[34–36]. Traditional electrolytes based on ethereal solvents and complexes from the reaction of organomagnesium Lewis bases and Lewis acids, such as organohaloaluminates, were able to operate with a thin passive layer to some extent; however, they were particularly prone to decomposition when exposed to moisture, necessitating a highly controlled environment for the manufacturing and storage[23]. Regarding more moisture-durable electrolytes such as ethereal solutions containing simple Mg salts (such as $Mg(TFSI)_2$ (TFSI = bis(trifluoromethanesulfonyl)imide) and $MgCl_2$), reversible Mg plating-stripping could be realized only in the absolute moisture-free environment[37]. More recently, alkoxyaluminate salts have emerged as highly moisture–resistant electrolytes, demonstrating stable Mg plating-stripping performance in the presence of up to 1000 ppm of water without a notable increase in overpotential[31,38,39]. While the moisture–resistance of the electrolyte system remains a critical challenge for the commercialization of RMBs, moisture scavenging in electrolytes is another widely employed strategy to address this issue. Organometallic reagents (e.g., $Bu_2Mg$, $Al_3Mg$, $Mg(BH_4)_2$, Mg powder) were often added to electrolytes to remove moisture and other impurities[40–43]. In most approaches, however, achieving good moisture–tolerance requires a rigorous drying procedure for all battery components as well as a strictly controlled manufacturing environment. These measures inevitably lead to a substantial increase in the battery manufacturing costs. For comparison, the cost associated with manufacturing Li-ion solid-state batteries that require a similar strict atmospheric control would be 150% higher than that of normal LIBs, even after the optimization[44]. Therefore, to ensure the commercial viability of rechargeable Mg batteries, it is utmost important to develop a methodology that ensures moisture tolerance at all steps of the battery manufacturing process. However, only a few notable studies have been reported on mitigating this challenge[32,33].

To address this critical issue, herein, a moisture-tolerant Mg electrode was proposed for effective application in the battery manufacturing process, even in a highly moist environment (~ 6500 ppm $H_2O$). This was accomplished by the formation of a sacrificial protective layer on the Mg metal via a simple surface modification that dispatched moisture scavengers into the electrolyte during the cell manufacturing process. The proposed method is highly scalable, and readily applicable to the existing battery manufacturing process. Therefore, this study provides a viable strategy for cost-effective manufacturing that will expedite the commercialization of rechargeable Mg batteries.

## Results

### Preparation of moisture-tolerant Mg electrodes via trimethyl phosphate treatment and Mg plating-stripping performance under the moist electrolytes

Reversible Mg plating-stripping in moisture-stable ethereal solutions based on simple Mg salts, such as $Mg(TFSI)_2/MgCl_2$ (0.5 M/0.5 M) in dimethoxyethane (G1) (denoted as E1), has usually been achieved in cases where the moisture content of the electrolytes was strictly controlled below a few dozen ppm[45]. This was demonstrated in Fig. 1a, where a Mg||Mg symmetric cell in a dry electrolyte containing 23 ppm water showed a decent plating-stripping behavior for hundreds of cycles with approximately 160 mV at a moderate current density of 0.1 mA cm⁻². However, the presence of small amount of water contaminant quickly passivated the Mg surface, triggering a catastrophic impedance growth (Fig. 1b), where moist electrolyte (374 ppm $H_2O$) prompted unimpeded large overpotential of several volts at the same current density due to the formation of ion-impermeable film on the surface[46]. This suggests that all battery components should be thoroughly dried before the cell assembly for Mg-ion batteries to function. For the electrolyte system, this may involve desiccating the Mg salts in a vacuum line at elevated temperatures and drying the solvents with

molecular sieves for several days before cell assembly. Moreover, a proper facility is required to restrain moisture inflow from the atmosphere throughout the cell manufacturing process. These cumbersome countermeasures are not only unfavorable for large-scale applications but also impose a sizable burden on manufacturing costs, creating a major roadblock to realizing practical Mg-ion batteries. To address these technical and cost challenges, a moisture-tolerant Mg electrode is proposed that can be effectively applied in battery manufacturing processes even in highly moist environments. The protected Mg electrode was prepared via a one-step solution dipping process, where the Mg metal was simply kept in trimethyl phosphate (TMP) for the optimized 15 min at room temperature (Fig. 1c, see Methods, and Supplementary Fig. 1–3 for details). This simple treatment enabled the Mg electrode to endure highly moist electrolytes containing a moisture content comparable to where undried salts were dissolved in the solvent (~ 1000 ppm $H_2O$). This suggests that the protected electrode can be applied to the cell manufacturing process in the absence of an atmosphere-controlling facility such as a glovebox, enabling fast and cost-effective battery production. Figure 1d, and e compared the Mg plating-stripping behaviors of the electrochemical cells with Mg||Mg symmetric configuration in the E1 electrolyte containing 1050 ppm water with untreated and the protected Mg electrodes, respectively. For the untreated Mg electrodes, an exceedingly large overpotential (~ 2 V) appeared from the incipient stage of Mg plating-stripping cycling. In contrast, the cell with the protected Mg electrode exhibited a reversible electrochemical behavior with a moderate overpotential of ~90 mV in 0.1 mA cm⁻², which is way smaller than that of untreated Mg electrodes in the electrolyte with a low moisture content (~ 160 mV from Fig. 1a). As shown in Supplementary Fig. 4, the protected Mg also exhibited a favorable moisture-tolerance in another ether-based electrolyte, E2: $Mg(TFSI)_2/MgCl_2$ (0.5 M/0.5 M) in diethylene glycol dimethyl ether (G2). This clearly indicates that the TMP treatment of Mg metal creates a common protection mechanism that endures high moisture in the electrolyte, facilitating facile $Mg^{2+}$ ion transport at the interface, whereas a compact ion-blocking layer forms on the surface without treatment.

The ultimate moisture tolerance of the TMP-treated electrode was evaluated by gradually increasing the moisture content in the electrolyte and measuring the reversibility of the Mg plating-stripping process. As shown in Fig. 2a, the overpotential profiles in 0.1 mA cm⁻² remained unaltered at approximately 100 mV until the moisture content reached 1050 ppm. It increased moderately to 180 mV at 6515 ppm, while its reversible character was firmly preserved. This is in stark contrast to the untreated electrode which functioned adequately only in the super-dried electrolytes (Fig. 1a, b). The electrode lost its tolerance apparently at 20000 ppm because a sharp increase in the overpotential to 2.0 V was observed from the incipient stage (Supplementary Fig. 5). This indicates that the protected electrode can maintain reversibility with a water content of at least thousands of ppm in the electrolytes, which would allow the electrode to endure the moisture uptake from the atmosphere during the battery manufacturing process in a typical dry-room facility. This capability is essential for economizing processing costs and enhancing productivity in battery manufacturing. The symmetric cell with the electrolyte containing 1050 ppm water managed a stable plating-stripping cycling for more than 1200 h with only a slight increase in its overpotential (Fig. 2b), while the reversibility was distinctly preserved for the electrolytes containing moisture as high as 6515 ppm (Supplementary Fig. 6). Figure 2c and Supplementary Fig. 7 exhibit the electrochemical behavior of the protected electrode at various current rates. The reversible character was well maintained up to 2 mA cm⁻² not only in the dry (23 ppm $H_2O$), but in the highly moist electrolytes (6515 ppm $H_2O$). The exchange current density, $i_o$, which is a useful indicator of the kinetics of the Mg plating-stripping reaction, was estimated for the electrolytes with various moisture contents. As in Fig. 2d, it was

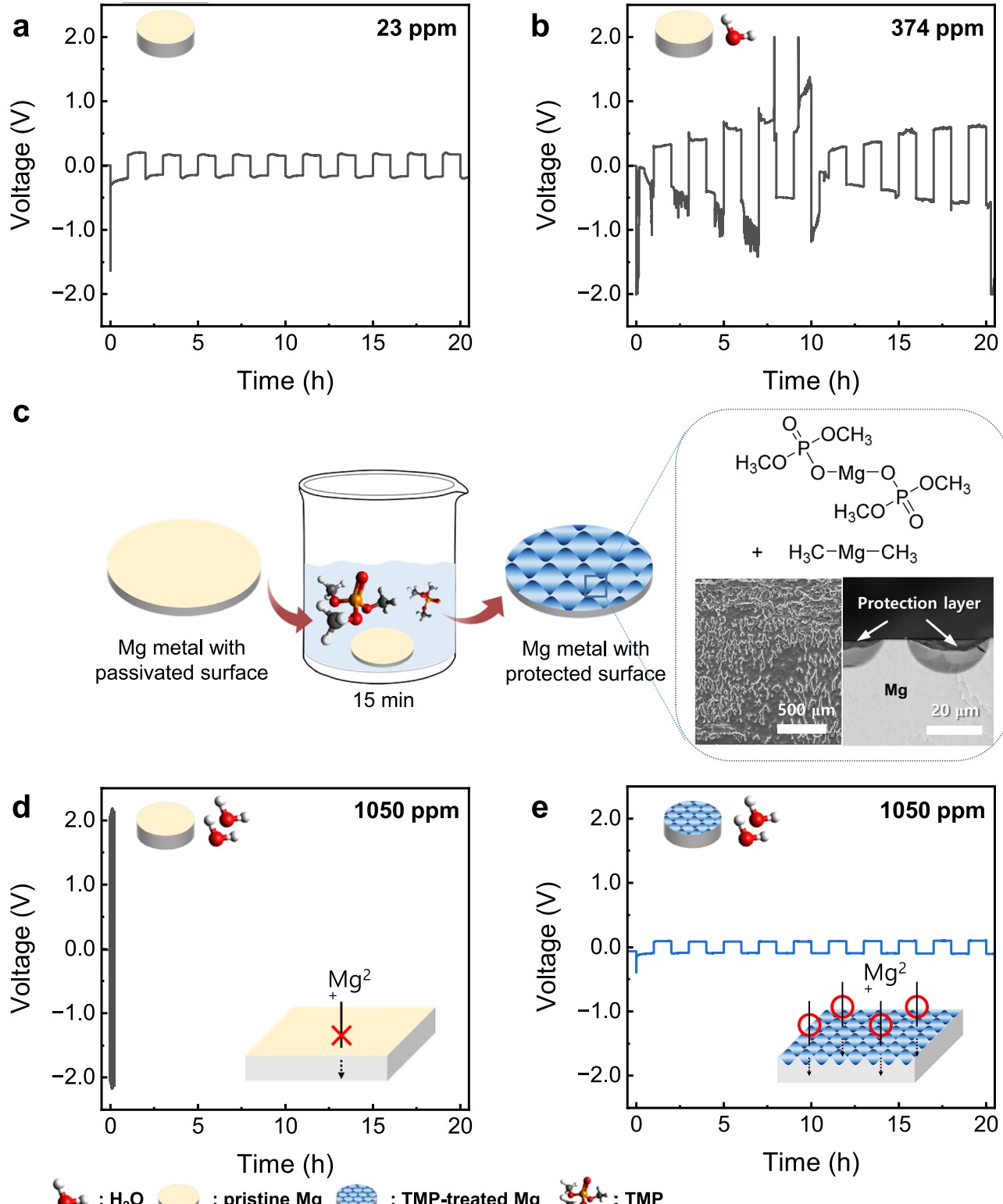

**Fig. 1 | Reversible Mg plating-stripping in highly moist electrolytes enabled by simple surface modification of Mg metal with trimethyl phosphate (TMP).** Typical Mg plating-stripping behavior of Mg||Mg symmetric cells with a pair of scraped Mg electrodes in **a** dry (23 ppm $H_2O$) and **b** moist (374 ppm $H_2O$) E1 electrolytes. **c** Schematic for preparing TMP-treated Mg electrodes with descriptive morphological features (top and cross-sectional view) and associated surface moieties. Mg plating-stripping behavior in a highly moist (1050 ppm $H_2O$) E1 electrolyte with **d** a pair of scraped (untreated) Mg electrodes and **e** a pair of TMP-treated Mg electrodes. The plating-stripping was performed under 0.1 mA cm$^{-2}$ and 0.1 mAh cm$^{-2}$ condition. The light yellow and blue disks represent the scraped pristine and TMP-treated Mg electrodes, respectively.

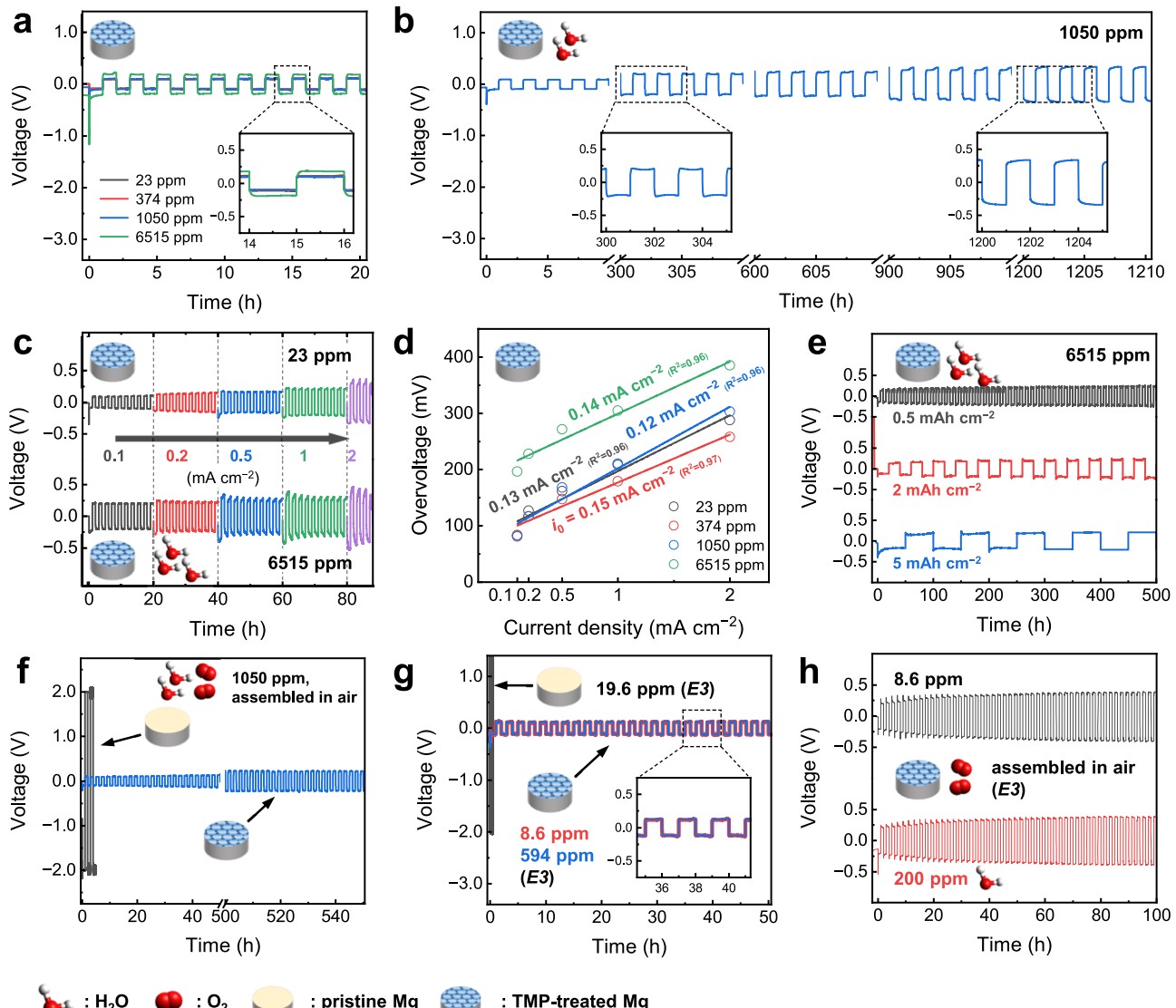

**Fig. 2 | Electrochemical characterization of trimethyl phosphate (TMP)-treated Mg electrodes under varying moisture conditions. a** Mg plating-stripping behavior of symmetric cells comprising a pair of TMP-treated Mg electrodes in E1 with varying moisture contents (23, 374, 1050, 6515 ppm). **b** Potential profile evolution during prolonged cycling in E1 with 1050 ppm $H_2O$. **c** Mg plating-stripping behavior under varying current rates (0.1, 0.2, 0.5, 1, 2 mA cm$^{-2}$) for E1 with 23 and 6515 ppm $H_2O$, and **d** the estimated exchange current densities ($i_o$) for each moisturized electrolyte with specified $H_2O$ content. **e** Mg plating-stripping behavior under varying Mg deposition quantities (0.5, 2, 5 mAh cm$^{-2}$) in highly moist (6515

ppm $H_2O$) E1. **f** Mg plating-stripping behavior of a cell composed of untreated Mg electrodes and a cell made of TMP-treated Mg electrodes in moist E1 (1050 ppm $H_2O$) assembled under atmospheric condition in dry-room. **g** Comparison of Mg plating-stripping behavior of a cell made of untreated Mg electrodes and a cell composed of TMP-treated Mg electrodes under dry or moist E3. **h** Mg plating-stripping behavior of the cells comprising TMP-treated Mg electrodes in a dry (8.6 ppm $H_2O$) or moist (200 ppm $H_2O$) E3 assembled under atmospheric conditions in a dry-room.

estimated to be between 0.12–0.15 mA cm$^{-2}$ regardless of moisture level in the electrolyte, indicating that the current moisture tolerance mechanism did not interfere with the electrode kinetics associated with Mg plating-stripping. The reversibility of Mg plating-stripping was also examined, depending on the amount of Mg metal deposited in the electrolytes at various moisture levels (Supplementary Fig. 8). A moderate reversibility was maintained up to 5 mAh cm$^{-2}$ even under a highly moist electrolyte (6515 ppm $H_2O$; Fig. 2e), corresponding to the state-of-the-art areal capacity density of commercial LIBs. These results confirm that TMP-treated Mg can be utilized as an effective moisture-tolerant negative metal electrode in Mg-ion batteries. The applicability of the protected electrode was further explored during the cell assembly process in an atmospheric environment in a dry-room facility. The symmetrical cell loaded with the protected electrodes in the moist electrolyte (1050 ppm) exhibited a reversible

behavior (Fig. 2f), whereas the cell with untreated Mg electrodes did not function properly, only exhibiting an excessive impedance growth from the beginning or short-circuited during the early cycles (Fig. 2f, Supplementary Fig. 9). This proves the applicability of the TMP-treated electrode to the manufacturing process in ambient atmosphere, which is one of the crucial factors in achieving good commercial–viability.

To investigate the practical applicability of this method in combination with oxide-type positive electrode materials, which are generally incompatible with cauterant halide-bearing electrolytes, such as dichloro-complex[47], all-phenyl-complex[22], and boron-centered anion-based magnesium electrolyte[48], TMP-treated Mg electrodes were examined with a halide-free electrolyte, such as E3: 0.5 M Mg(TFSI)$_2$ in G2. As shown in Fig. 2g, Mg plating-stripping with untreated Mg electrodes in E3 was highly irreversible, even with a dry electrolyte (19.6 ppm $H_2O$). However, reversible characteristics was readily achieved

with the TMP-treated Mg electrodes even in the highly moist electrolyte (4500 ppm $H_2O$; Fig. 2g, Supplementary Fig. 10). The reversible behavior in the electrolyte with mild moisture content (60 ppm) was well maintained for hundreds of hours (Supplementary Fig. 11). Moreover, stable performance at both high current rate and high areal capacity was observed under both dry (4 ppm $H_2O$) and moist (200 ppm $H_2O$) conditions (Supplementary Fig. 12 and Supplementary Fig. 13). The applicability of the protected electrode was also evaluated in the moist electrolyte in the ambient atmospheric condition in a dry-room facility. As shown in Fig. 2h, the reversible character was readily verified in a moist electrolyte containing 200 ppm $H_2O$. These results indicate that using the method can achieve high energy density in Mg-ion batteries by employing oxide-type positive electrode materials that may possess higher redox potentials and larger theoretical capacities than the heavily investigated sulfide-based materials[49].

The Mg plating-stripping behavior was also evaluated under an asymmetrical cell configuration. The Mg||stainless steel (SS) cell with a protected Mg electrode exhibited highly reversible characteristics, with an efficiency of ~98% in a moist E1 electrolyte containing 1050 ppm $H_2O$ (Supplementary Fig. 14), whereas that of untreated Mg did not function at all (Supplementary Fig. 15). Moreover, Mg metal crystals were clearly observed in the deposits from the moist electrolytes in the Mg||Cu cells, regardless of the moisture content in E1 (Supplementary Fig. 16) and E3 (Supplementary Fig. 17). This implied that the high moisture content in the electrolyte was quickly eliminated by the built-in moisture tolerance mechanism in the TMP-treated Mg electrode.

## Origin of moisture tolerance of TMP-treated Mg electrodes

To understand the origin of the unique moisture tolerance of the TMP-treated Mg electrode, its surface moieties were characterized using Fourier-transform infrared (FT-IR) spectroscopy (Fig. 3a). The spectrum of the TMP-treated electrode was similar to that of Mg dimethyl phosphate ($Mg(DMP)_2$) but differed from that of TMP, indicating that the surface state of the Mg electrode was altered via a chemical reaction between the Mg electrode and TMP. This is consistent with a previous study on the chemical reactions of Mg metal in TMP, which proposed $Mg(DMP)_2$ as the principal reaction product, along with dimethylmagnesium ($Me_2Mg$)[50]. To further clarify the nature of the changes in the Mg electrode, a TMP-treated Mg electrode was immersed in deionized water and the gas components that evolved subsequently were analyzed using gas chromatography-flame ionization detector (GC-FID) (Fig. 3b). When $Me_2Mg$ is one of the products of the chemical reaction between the Mg electrode and TMP, it will likely be converted into methane ($CH_4$) and magnesium hydroxide via the following chemical reaction:

$$Me_2Mg + 2H_2O \rightarrow Mg(OH)_2 + 2CH_4(g) \qquad (1)$$

In the GC-FID spectra, $CH_4$ gas was distinctly observed when TMP-treated Mg metal was placed in water, verifying the presence of $Me_2Mg$. The FT-IR spectra of the dried THF or G1 solution[51], collected after 10 min. Immersion of TMP-treated Mg and subsequent filtration, revealed the presence of $Me_2Mg$ in the protective layer (Supplementary Figs. 18–21). Moreover, when $Mg(DMP)_2$ or untreated Mg metal was placed in pure water, no gas evolution was detected, indicating that $Me_2Mg$ was exclusively responsible for methane evolution. A substantial increase in the $CH_4$ evolution (~500%) was observed when water was added to the TMP solution, where Mg metal was completely dissolved, as it should contain more $Me_2Mg$. The small amount of $C_2H_6$ gas detected may have arisen from the combination of methyl radical intermediates during the reaction[50]. Besides, no intrinsic decomposition products were observed in the $^1H$ NMR spectra, even when $Mg(DMP)_2$ was exposed to $H_2O$ (Supplementary Fig. 22), further

confirming that the methane did not stem from $Mg(DMP)_2$. As anticipated from reaction (1), the white precipitate collected from these reactions was identified as magnesium hydroxide ($Mg(OH)_2$) from the XRD analysis (Fig. 3c). These results suggest that $Me_2Mg$ is most probably the origin of the moisture tolerance of the TMP-treated Mg electrode.

To further characterize the surface structure, X-ray absorption spectroscopy (XAS) and X-ray photoelectron spectroscopy (XPS) were performed on the TMP-treated Mg electrode and associated materials. The Mg K-edge absorption spectra indicated the formation of a thin surface layer on Mg metal that differed in nature from that of a normal passive film of Mg metal (Supplementary Fig. 23). The XPS P 2$p$ peak at 134.1 eV observed in the TMP-treated Mg (Fig. 3d) confirms the presence of a phosphate-based surface film originating from the chemical reaction between TMP and Mg metal. Moreover, a slight shift (0.49 eV) in the Mg 2$p$ peak for TMP-treated Mg to a lower binding energy than that of $Mg(DMP)_2$ may be associated with the $Me_2Mg$ component in the protective layer.

Based on the observations in this study, a plausible mechanism for the moisture tolerance of the TMP-treated Mg electrode is proposed in Fig. 3e. In the as-prepared TMP-treated Mg electrode, the surface was probably covered with a composite film of $Me_2Mg$ in the $Mg(DMP)_2$ matrix. During the cell manufacturing process, the $Me_2Mg$ in this film may readily permeate the electrolyte, scavenging $H_2O$ molecules in the electrolyte or from the air and protecting the vulnerable surface from the moisture contamination. The resulting byproduct, $Mg(OH)_2$, is believed to remain suspended in the electrolyte or become adsorbed onto the nearby separator (Supplementary Table 1). Moreover, because the TMP-treated Mg is in intimate contact with the moist electrolyte, $H_2O$ molecules may be drawn and adsorbed onto the hydrophilic phosphate moieties in $Mg(DMP)_2$ (Supplementary Fig. 24, Supplementary Data 1)[52], further reacting with $Me_2Mg$ to form $CH_4$ and $Mg(OH)_2$. This simultaneous chemical scavenging and physical trapping of water molecules synergistically dehydrates the electrolyte, lowering the moisture content in the electrolyte rapidly enough to enable reversible Mg plating-stripping. This also explains why TMP-treated Mg electrodes worked efficiently in Mg plating-stripping even with an asymmetrical configuration (Supplementary Fig. 14). The proposed scavenging mechanism was further supported by direct measurements of moisture concentration after TMP-treated Mg metals were immersed in the moist electrolyte, revealing a significant decrease in moisture level within 3 h (Supplementary Table 2). In addition, moisture-scavenging by $Me_2Mg$ in the moist electrolyte is considered fast and highly effective, as reversible Mg plating-stripping was observed even without an initial resting period (Supplementary Fig. 25). Notably, the protective layers on Mg metal remained relatively intact throughout electrochemical cycling, implying that they may facilitate uniform Mg deposition in addition to their role as moisture-scavengers during cell assembly (Supplementary Figs. 26–28).

## Battery performance of Mg-ion cells loaded with TMP-treated Mg electrodes in moist electrolytes

To demonstrate the applicability of the TMP-treated Mg electrode in actual Mg-ion batteries under varying conditions, several positive electrode materials were employed to configure full cells. Firstly, the performance of the $Mo_6S_8$ Chevrel phase (Supplementary Fig. 29), a representative insertion host for multivalent cations[47], was evaluated at 12 mA $g^{-1}$ (~1/10 C) with the E1 electrolyte. Supplementary Fig. 30 and Fig. 4a show the electrochemical behavior of battery cells manufactured in an argon-filled glove box with dry (23 ppm $H_2O$) and moist (374 ppm $H_2O$) electrolytes, respectively. They shared almost identical voltage profiles and exhibited stable cycle performances, delivering reversible capacities at approximately 70 mAh $g^{-1}$ regardless of the moisture content in the electrolyte. Moreover, the battery cell manufactured in an atmospheric environment (without the aid of a

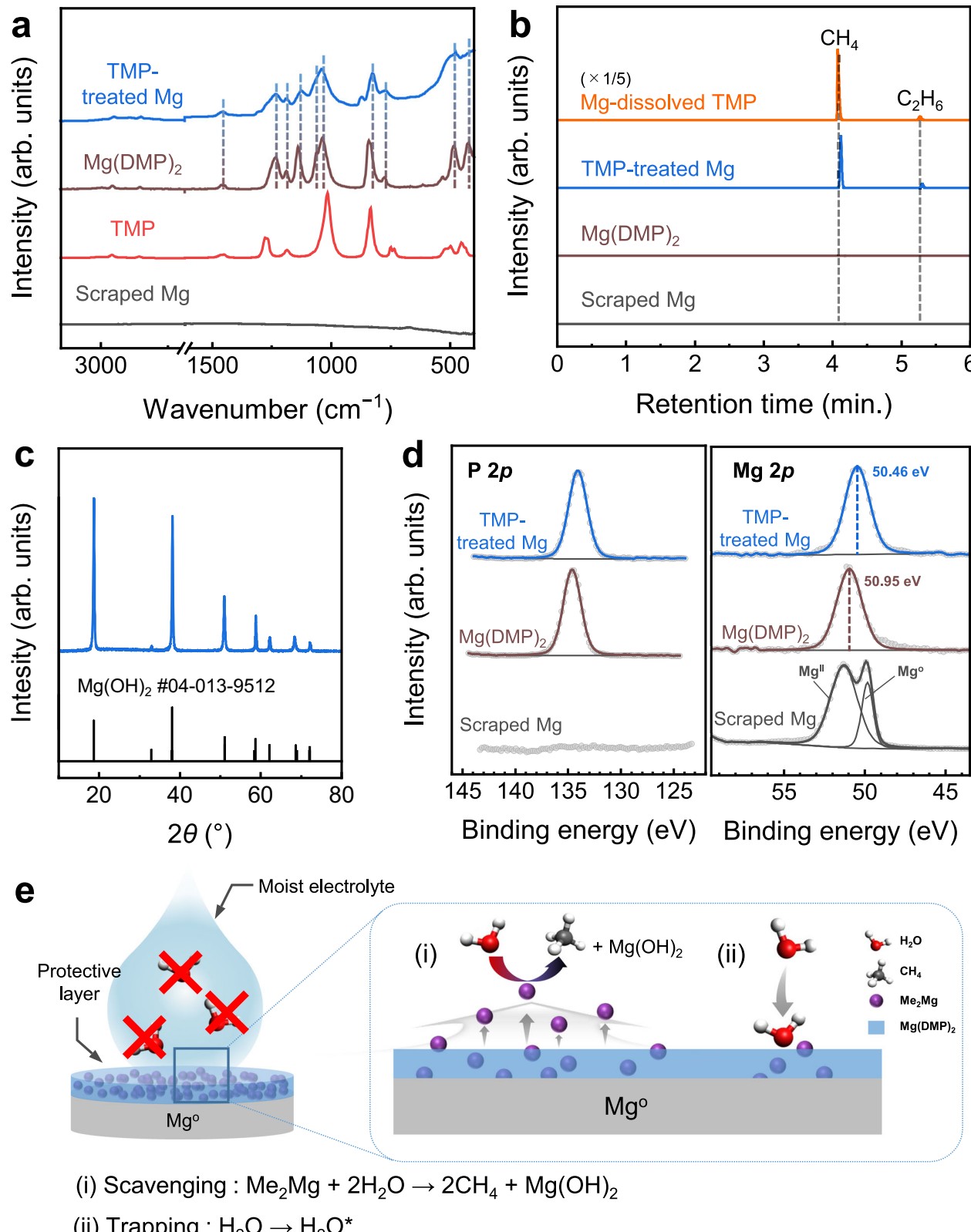

(i) Scavenging : $Me_2Mg + 2H_2O \rightarrow 2CH_4 + Mg(OH)_2$

(ii) Trapping : $H_2O \rightarrow H_2O^*$

glovebox) maintained stable electrochemical performance for hundreds of cycles (Fig. 4b), indicating that the moisture-resistance mechanism effectively mitigates the impact of moisture exposure during cell manufacturing. Comparable battery performance was also observed in $Mo_6S_8$ with the E3 electrolyte (Supplementary Fig. 31). These findings suggest that the protected electrode can be processed in manufacturing environments similar to those of current Li-ion

batteries, enabling the production of Mg-ion batteries at significantly lower cost. The protected Mg electrode was also combined with an ammonium vanadate ($NH_4V_4O_{10}$, NVO) positive electrode material (Supplementary Fig. 32) and a halide-free E3 electrolyte with some moisture content. NVO is known to possess a layered crystal structure with a wide interlayer spacing (~10 Å) that could favor the incorporation of multivalent cations[53]. However, it degraded rapidly in

**Fig. 3 | Characterization of the surface moieties on the trimethyl phosphate (TMP)-treated Mg electrode and proposed moisture tolerance mechanism.** **a** FT-IR spectra of the TMP-treated Mg electrode and associated materials (TMP, Mg(DMP)$_2$, and scraped Mg). **b** GC-FID spectra for the gases that evolved after the TMP-treated Mg electrode or associated materials were dipped in deionized water. Mg-dissolved TMP describes the TMP solution where Mg reacted with TMP for one day. **c** X-ray diffraction pattern of the white precipitate recovered from the reaction between TMP-treated Mg and pure water. **d** X-ray photoelectron spectroscopy P 2$p$ and Mg 2$p$ spectra of TMP-treated Mg electrode, Mg(DMP)$_2$, and scraped Mg foil. **e** Proposed moisture tolerance mechanism of TMP-treated Mg electrode. During cell manufacturing, the Me$_2$Mg component in the surface film of the TMP-treated Mg electrode readily permeates into the ether-based electrolyte to eliminate any water molecule that it encounters. At the same time, the hydrophilic phosphate component traps water molecules to inhibit the infiltration of moisture onto the bare Mg surface. H$_2$O* in (**e**) represents the H$_2$O adsorbed on the protective layer.

typical halide-bearing ethereal electrolytes due to prompt corrosion. Reportedly, NVO delivered a reversible capacity of approximately 180 mAh g$^{-1}$ upon intercalation with Mg$^{2+}$ in mild electrolyte systems such as Mg(ClO$_4$)$_2$ in acetonitrile. However, acetonitrile-based electrolytes are incompatible with Mg metal electrodes[54]. The protected Mg electrode enabled the direct use of oxide-type positive electrodes and Mg metal electrodes, in combination with an ethereal E3 electrolyte, to form full cells. Figure 4c, d show the voltage profiles of the NVO||Mg cells in an E3 electrolyte under dry (20 ppm H$_2$O) and moist (60 ppm H$_2$O) conditions, respectively. The battery cells assembled in air with a moist electrolyte function adequately, moderately preserving the original voltage profiles even after prolonged cycling. This result demonstrates that oxide-type materials can be safely used in combination with protected Mg electrodes to form high-energy-density full cells. Thirdly, the TMP-treated Mg electrode was paired with activated carbon cloths (ACCs) (Supplementary Fig. 33) to construct a dual-ion battery based on the E3 electrolyte. In many carbonaceous materials, electrical charges can be stored via surface adsorption or intercalation of anions in the graphene layers[55]. Figure 4e and f show the voltage profiles of ACC in dry (20 ppm) and moist (220 ppm) electrolytes, respectively. Typical capacitive profiles for dozens of cycles with a favorable reversibility were observed in both cases. These examples show that TMP-treated Mg can serve as a ubiquitous negative electrode for constructing Mg-ion batteries under various conditions.

## Discussion

To enable practical dry-room manufacturing of RMBs using the strategy developed in this work, it is essential to secure a sufficient protection time window for cell assembly, given the high reactivity of Me$_2$Mg with ambient moisture and oxygen. To this end, the reversibility of Mg plating-stripping was evaluated in moist electrolyte (997 ppm H$_2$O) with TMP-treated Mg that had been exposed to dry room air for various durations. As shown in Fig. 5a, the reversibility was relatively well maintained up to 1 h of air exposure. However, after 3 h, moisture-tolerance was clearly lost, as evidenced by a sharp increase in the overpotential during Mg plating-stripping. GC-FID analysis of the gases that evolved from the reaction between the moist electrolyte and TMP-treated Mg, which was pre-exposed to dry-room air for different durations, revealed that a substantial amount of CH$_4$ was still generated after 30 min of air exposure (Fig. 5b, Supplementary Fig. 34). These results indicate that the scavenging agent, Me$_2$Mg, not only removed moisture from the electrolyte but also continuously eliminated moisture and oxygen from the environment during the entire manufacturing process. This highlights the potential of our strategy to significantly improve the feasibility of dry-room manufacturing of RMBs.

To assess the contribution of physical scavenging to the overall dehydration process, we examined the role of the Mg(DMP)$_2$ layer formed on the TMP-treated Mg surface. This layer can adsorb residual moisture from the electrolyte, as supported by a separate test showing that synthesized Mg(DMP)$_2$ powder reduced the H$_2$O content of a moist electrolyte by approximately 660 ppm (Supplementary Table 3). These results indicate that the Mg(DMP)$_2$ layer provides an auxiliary pathway for moisture removal, complementing the chemical scavenging of Me$_2$Mg.

To further confirm that the electrochemical process observed in moist electrolytes corresponds to Mg plating and stripping, additional three-electrode measurements were carried out using a modified Ag/AgCl reference electrode[47,56]. The cyclic voltammograms (Fig. 5c, d, Supplementary Fig. 35) clearly showed reversible Mg plating-stripping peaks even in moist electrolytes when TMP-treated Mg was used as the counter electrode, while no additional peaks attributable to hydrogen evolution or oxidation reactions were detected. Notably, the Mg/Mg$^{2+}$ redox potential remained nearly identical between dry and moist electrolytes (around −2.2 V vs. Ag/AgCl), indicating that the electrochemical process was not influenced by moisture-induced side reactions. These results confirm that the current response in the moist electrolyte originates from Mg plating-stripping rather than from side reactions, thereby supporting the proposed mechanism.

Electrolytes based on weakly-coordinated anions, such as ethereal solutions of Mg alkoxyborates and alkoxyaluminates, have attracted considerable research interest in recent years[31,38,39]. In particular, Mg[Al(hfip)$_4$]$_2$ in G2 (hfip = 1,1,1,3,3,3-hexafluoroisopropyloxy) has demonstrated efficient Mg plating-stripping performance even in the presence of 1000 ppm of water, without a significant increase in overpotential[39]. To assess the compatibility of our strategy with such electrolytes, the reversibility of Mg plating-stripping was investigated in E4 electrolyte: 0.3 M Mg[B(hfip)$_4$]$_2$ in G1, containing moisture levels of 2.0, 1092, and 3482 ppm H$_2$O, respectively – a system whose moisture tolerance has been relatively unexplored. With TMP-treated Mg electrodes, reversible Mg plating-stripping was still achievable even under highly humid conditions (up to ~3500 ppm H$_2$O), although a moderate increase in overpotential was observed (Supplementary Fig. 36). This likely arises from thin passivation films formed by residual moisture, whereas chloride-containing electrolytes showed better moisture tolerance and low overpotential due to chloride-induced disruption of these films. These results demonstrate the versatility of our strategy across various electrolyte systems.

It is noteworthy that TMP has often been reported as an additive or co-solvent of the electrolyte for Mg-ion batteries[19,57]. These studies primarily focused on either modifying the solvation structure by incorporating TMP molecules in the solvation sheath, or influencing interfacial chemistry by constructing ion-conductive inorganic SEI layers rich in fluorides and phosphides. However, aspects such as a moisture-tolerant manufacturing process for RMBs were not seriously addressed in these studies. In contrast, the present work focuses on developing a viable strategy to secure moisture tolerance through a simple and convenient surface activation process in TMP, thereby enabling cost-effective dry-room manufacturing of RMBs.

In summary, a viable strategy for constructing moisture-tolerant RMBs was established by introducing a protective surface layer on Mg metal. This approach utilizes moisture-sensitive moieties on Mg metal that act as efficient moisture-scavengers and moisture-traps during the battery manufacturing process. This concept was realized through a simple surface modification using TMP, imparting sufficient moisture-tolerance to allow Mg metal to withstand highly moist and/or atmospheric manufacturing conditions. This work addresses one of the critical challenges in the commercialization of RMBs.

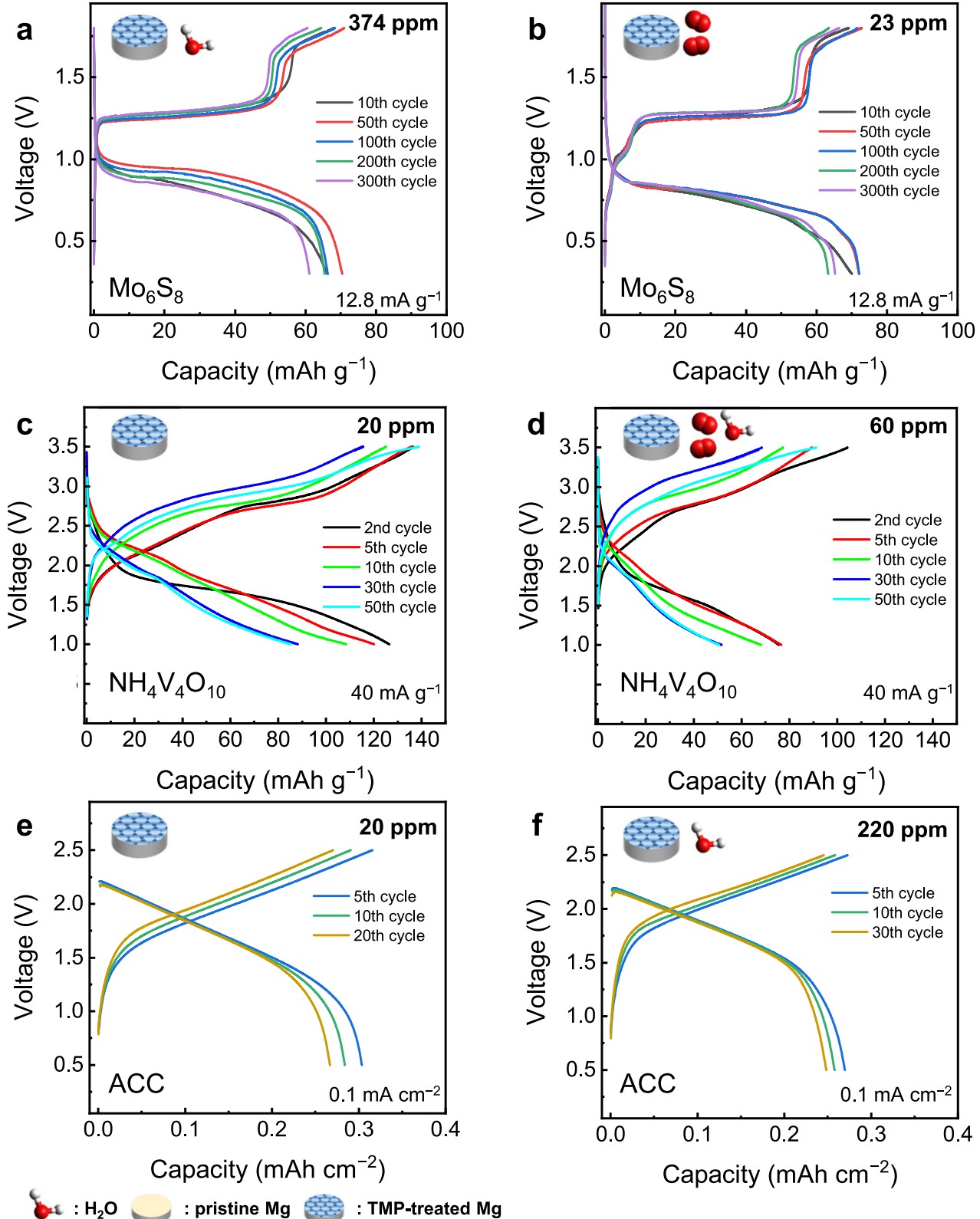

**: H₂O**   **: pristine Mg**   **: TMP-treated Mg**

## Methods

### Preparation of TMP treated Mg electrode

Mg foil (HanaroTR, 99.9%, 0.1 mm thick) was scraped with a scalpel blade to remove the native passivation layer on the Mg surface. Subsequently, the scraped Mg metal was punched into a disk with a diameter of 16 mm (untreated Mg electrode) and immersed in 1 mL of dried trimethyl phosphate (TMP) (Sigma-Aldrich, ≥99%) for 15 min to form a uniform protective film. The excessive TMP on the Mg surface was wiped off with dried disposable wipers (Kimwipes, Yuhan-Kimberly). The entire process of preparing TMP-treated Mg was performed in an argon-filled glove box ($H_2O < 0.1$, $O_2 < 0.1$ ppm) to prevent the formation of any dispensable passive film. TMP was dried with molecular sieves (4 Å, Daejung), which were activated at 200 °C for 24 h under a vacuum line just before use.

**Fig. 4 | Electrochemical performance of full cells with a trimethyl phosphate (TMP)-treated Mg metal negative electrode and various feasible positive electrodes. a** The discharge-charge profiles of the cell composed of the $Mo_6S_8$ Chevrel phase positive electrode and TMP-treated Mg metal negative electrode in the moist E1 (374 ppm $H_2O$). **b** The discharge-charge profiles of the cell composed of the $Mo_6S_8$ and TMP-treated Mg metal assembled under atmospheric conditions in a dry room. **c** The discharge-charge profiles of the cell composed of the $NH_4V_4O_{10}$ positive electrode and TMP-treated Mg metal negative electrode in a chloride-free E3 electrolyte. **d** The discharge-charge profiles of the cell composed of the $NH_4V_4O_{10}$ and TMP-treated Mg metal in the moist E3 (60 ppm $H_2O$) assembled under atmospheric conditions in a dry room. **e** The discharge-charge profiles of the cell composed of the activated carbon cloth (ACC) positive electrode and TMP-treated Mg metal negative electrode in the E3. **f** The discharge-charge profiles of the cell composed of ACC and TMP-treated Mg metal in the moist E3 electrolyte (220 ppm $H_2O$). The typical mass loading of active materials for (**a**–**d**) was ~1.5 mg cm⁻². For ACC, areal density was ~ 9 mg cm⁻².

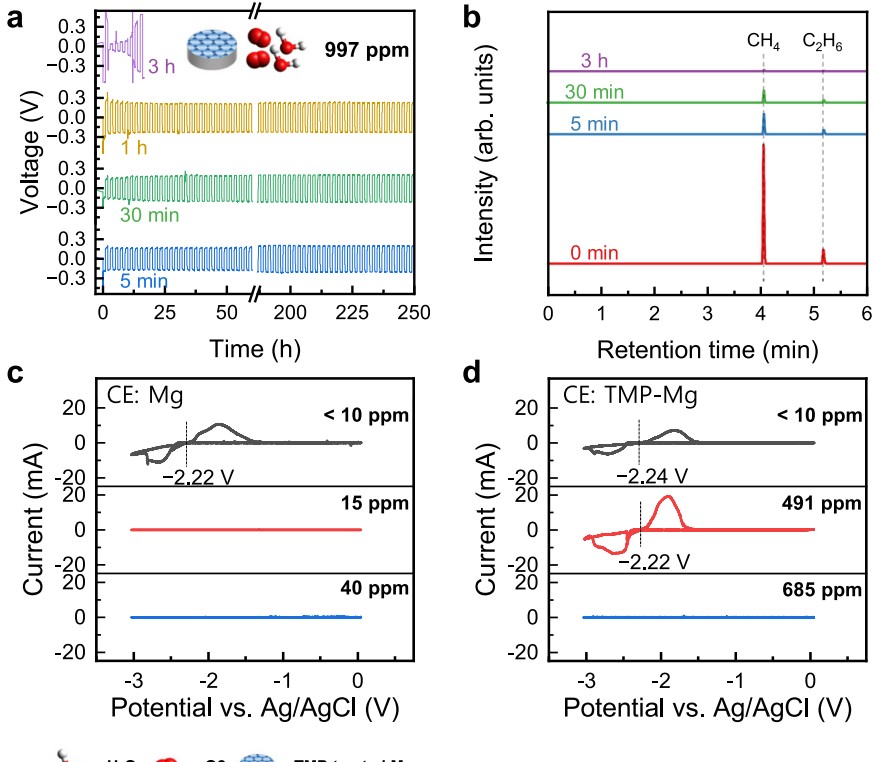

**Fig. 5 | Characterization of trimethyl phosphate (TMP)-treated Mg electrodes pre-exposed to dry-room air for different time period and verification of Mg redox potentials via cyclic voltammetry. a** Mg plating-stripping behavior of symmetrical cells in E1 (997 ppm $H_2O$) composed of a pair of TMP-treated Mg electrodes, which was pre-exposed to dry room air for 5 min, 30 min, 1 h, and 3 h. **b** Gas chromatography-flame ionization detector spectra for the gases that evolved after the TMP-treated Mg electrode, which were pre-exposed to dry room air for 0 min, 5 min, 30 min, and 3 h, respectively, were dipped in moist electrolyte (E1, 997 ppm $H_2O$). Cyclic voltammetric curves obtained from three-electrode cells with Ag/AgCl as the reference electrode, **c** a scraped pristine Mg disk as the counter electrode (CE), and **d** a TMP-treated Mg disk as the CE in E1 electrolyte containing various water contents. The moisture tolerance in the three-electrode setup is about one-eighth that of the Mg‖Mg coin cell (see Methods for details).

## Preparation of electrolytes

To prepare E1, 2.5 mmol of magnesium bis(trifluoromethanesulfonyl) imide (Mg(TFSI)₂) (Solvionic, 99.5%, 1.462 g) powders were dissolved in dimethoxyethane (G1) (Sigma-Aldrich, 99.5%), and the solution was stirred at 80 °C for 1 h. Then, 2.5 mmol of MgCl₂ (Alfa Aesar, 99.99%, 0.238 g) was added to the solution to produce Mg(TFSI)₂/MgCl₂ (0.5 M/0.5 M) in G1 (5 mL). Mg(TFSI)₂ was dried at 200 °C for 24 h under a vacuum line just before use, and MgCl₂ was used as received. G1 were dried for at least three days by activated molecular sieves prior to electrolyte preparation. The moisture in the electrolyte was controlled by adjusting drying period of salts and solvents or adding additional deionized water. The moisture content in the electrolyte was measured in a dry room facility using Karl Fischer titration (899 Coulometer, Metrohm). E2 and E3 were prepared using diethylene glycol dimethyl ether (G2) (Sigma-Aldrich, 99.5%) by following the similar procedure. For E4, magnesium tetrakis(hexa-fluoroisopropyloxy)borate (Mg[B(hfip)₄]₂) was dissolved in G1 (0.4957 g in 0.62 mL) to prepare a 0.3 M solution.

## Synthesis of Mg[B(hfip)₄]₂

Under an argon atmosphere, a solution of di-n-butyl magnesium (1 M in heptane, 25 mL, 25 mmol, 1 eq.; Sigma-Aldrich, Quality level 100) was cooled down to around −20 °C. Then hexafluoroisopropanol (5.4 mL, 8.6 g, 51 mmol, 2.05 eq.; abcr, 99%), previously dried over activated molecular sieve (4 Å) for at least one night prior to use, was added dropwise. The reaction mixture was heated to 25 °C and stirred for 1.5 h. Dimethoxyethane (50 mL) was then added to dissolve the precipitated magnesium hexafluoroisopropoxide. The resulting solution was cooled again to approximately −50 °C, followed by dropwise addition of BH₃•THF (1 M in THF, 55 mL, 55 mmol, 2.2 eq.; Thermo Scientific Acros, stabilized). Subsequently, additional hexafluoroisopropanol (17.9 mL, 28.6 g, 170 mmol, 6.8 eq.) was added dropwise. The reaction mixture was allowed to reach 25 °C overnight and then stirred for an additional 2 h. Afterwards, the solvents were removed under reduced pressure (~10⁻² mbar) with a gradual heating to 60 °C. The resulting white solid was dried under vacuum at 60 °C overnight. The white material was then transferred into a BÜCHI tube

inside an argon-filled glovebox and dried under vacuum at 60 °C for an additional night. Finally, Mg[B(hfip)$_4$]$_2$ • 3DME (35 g, 21 mmol, 84%) was obtained as a white crystalline solid.

## Synthesis of Chevrel phase (Mo$_6$S$_8$)

Firstly, Cu$_2$Mo$_6$S$_8$ was synthesized via previously reported molten salt method[58,59] with some modifications. Briefly, stoichiometric amounts of Mo (Sigma-Aldrich, 99.9%), MoS$_2$ (Sigma-Aldrich, 98%), and CuS powders (Sigma-Aldrich, ≥99%) were mixed with dried KCl (Sigma-Aldrich) at a KCl-to-mixture weight ratio of 2. The mixture was heated at a rate of 2 °C min$^{-1}$ and calcined at 850 °C for 60 h under a flowing argon atmosphere. Then, KCl was washed out with plenty of deionized water from the product to obtain Cu$_2$Mo$_6$S$_8$. To eliminate Cu component, the product was oxidized via stirring in 6 M HCl solution (Samchun) under O$_2$ blowing condition for 3 days. The resulting powder (Mo$_6$S$_8$) was filtered, washed several times with an excess amount of deionized water, and dried in the oven at 80 °C overnight.

## Synthesis of NH$_4$V$_4$O$_{10}$

NH$_4$V$_4$O$_{10}$ material was synthesized through a hydrothermal method, which was adapted from a previously reported method[60]. Briefly, 10 mmol of NH$_4$VO$_3$ (Sigma-Aldrich, 1.170 g) was dissolved in 80 °C deionized water to form a light-yellow solution. 15 mmol of H$_2$C$_2$O$_4$·2H$_2$O (Daejung, 1.891 g) powder was then added to the solution while maintaining stirring at 80 °C until it became a dark-green solution. The resulting solution was transferred to a 150 ml autoclave and maintained in an oven at 140 °C for 48 h. After being cooled to room temperature, the products were washed with plenty of deionized water, and dried at 80 °C overnight to obtain NH$_4$V$_4$O$_{10}$.

## Activated carbon cloth (ACC)

Activated carbon cloth (ACC) labeled ACC-507-20 (dimensions: mass of 90 g m$^{-2}$, thickness of 0.43 mm, surface area of >1800 m$^2$ g$^{-1}$) was purchased from Kynol Europa GmbH. ACC was dried at 200 °C for 24 h under a vacuum line and then cut into a disk with a diameter of 16 mm in the glovebox.

## Synthesis of Mg dimethyl phosphate

To synthesize Mg dimethyl phosphate, Mg(DMP)$_2$, 4 mmol of TMP (0.43 mL) was mixed with 2 mmol of MgI$_2$ (0.507 g) (Sigma-Aldrich, 98%) in G1 (20 mL) by following a previously reported synthesis of NaDMP[61]. The solution was stirred at 50 °C for 3 days in an Ar-filled glove box. Then, the resulting product was filtered and dried under vacuum. The FT-IR spectrum of Mg(DMP)$_2$ was similar to that of NaDMP prepared by the previously reported method, confirming the successful synthesis of Mg(DMP)$_2$ (Supplementary Fig. 37).

## Electrochemical evaluation using two-electrode cells

Mg plating-stripping behavior was investigated using CR2032-type coin cells (316 stainless steel) with a symmetric (Mg||Mg), or an asymmetric (Mg||stainless steel (SS)) configuration. The cells were assembled in an argon-filled glovebox (H$_2$O < 0.1, O$_2$ < 0.1 ppm) or under an atmosphere in the dry room facility, where due point was maintained below −60 °C. A glass fiber (GF/F, 0.42 mm thick, Whatman) was punched into disks with a diameter of 18 mm and utilized as the separator. Ether-based electrolytes (100 μL) with various moisture contents were applied to the cells with one piece of glass fiber separator to evaluate the moisture tolerance of TMP-treated Mg electrodes. The cells were put to a resting period of 3 h before the current was applied unless specified otherwise. Mg plating-stripping was carried out with 0.1 mA cm$^{-2}$, and 0.1 mAh cm$^{-2}$ condition, unless specified otherwise. The asymmetric cells were tested without any pre-cycling procedure. For the full cell evaluation, the positive electrodes were prepared as follows. 80 mg of electrode material (Mo$_6$S$_8$ or NH$_4$V$_4$O$_{10}$), 10 mg of Super P, and 10 mg of poly(vinylidene fluoride) (PVdF) (Solvay, Solef 5130) were mixed in *N*-methyl-

2-pyrrolidone (NMP) (Sigma-Aldrich, 99.5%) via ball milling (Mini-Mill Pulverisette 23, Fritsch) at 40 Hz for 30 min. The slurry (solvent-to-solid ratio = 7) was cast onto one side of a Ti foil (Alfa Aesar, 0.032 mm thick, 99.7%) using a doctor blade and dried at 80 °C overnight under vacuum. The typical mass loading of active material was ~1.5 mg cm$^{-2}$ for both Mo$_6$S$_8$ and NH$_4$V$_4$O$_{10}$. The dried coated foil was then punched into disks with a diameter of 12 mm. The coin-type full cells with various positive electrode materials were assembled in an argon-filled glovebox or in a dry room under an atmosphere. All electrochemical tests were performed in galvanostatic mode using a Maccor Series 4000 battery tester in a climatic chamber maintained at 25 ± 0.5 °C. The Mg||Mo$_6$S$_8$ cells (1 C = 128 mAh g$^{-1}$) were cycled at 0.1 C in the voltage range of 0.3 and 1.8 V, while Mg||NH$_4$V$_4$O$_{10}$ cells (1 C = 400 mA g$^{-1}$) were cycled at 0.1 C in the voltage range of 1.0 and 3.5 V. For dual-ion Mg||ACC cells, they were cycled at 0.1 mA cm$^{-2}$ in the voltage range of 0.5 and 2.5 V. No pre-activation procedure was applied to the negative electrode prior to full-cell testing. Unless otherwise stated, each electrochemical experiment was repeated using at least two independent cells per condition.

## Electrochemical evaluation using three-electrode cells

To verify that the voltage profile of the TMP-treated Mg electrode originates primarily from Mg plating behavior rather than the hydrogen evolution reaction caused by residual moisture in the electrolyte, an electrochemical evaluation was conducted using a three-electrode configuration. The cell consisted of polytetrafluoroethylene (PTFE) Tee–union connector body sealed with PFPE ferrules and stainless steel plungers. A stainless steel disk (diameter: 12 mm) and an Mg or TMP-treated Mg metal disk (diameter: 12 mm) were used as the working and counter electrodes, respectively. A commercial Ag/AgCl electrode immersed in dried E1 served as the reference electrode. Two sheets of glass fiber separators were employed, and 200 μL of E1 electrolyte was added to the three-electrode cell. Cyclic voltammetry (CV) was conducted using a potentiostat (VMP-3, Bio-logic) with cut-off voltages between −3.0 and 0.25 V vs. Ag/AgCl at a scan rate of 10 mV s$^{-1}$. Prior to the CV measurement, the assembled cell was rested for 3 h under open-circuit conditions, consistent with the procedure used for two-electrode coin cells. Scaling note. Because the three-electrode cell uses twice the electrolyte volume (0.2 vs 0.1 mL), roughly half the electrode area (1.1 vs 2.0 cm$^2$), and only one TMP-treated Mg electrode (vs. two in the Mg||Mg coin cell), its effective moisture tolerance is ≈1/8 that of the coin cell (scaling ≈ (0.1/0.2) × (1.1/2.0) × (1/2) ≈ 0.14). Hence, 100 ppm H$_2$O in the three-electrode cell corresponds to ~ 800 ppm in the Mg||Mg coin cell.

## Material characterization

Surficial and cross-sectional morphology of the electrode was observed using a field-emission scanning electron microscope (FE-SEM; Regulus 8230, Hitachi) equipped with energy dispersive X-ray spectroscopy (EDS) operating at an accelerating voltage of 15 kV. Ion milling (IMS, ArBlade 5000, Hitachi) was used to prepare specimens for cross-sectional image. X-ray photoelectron spectroscopy (XPS; PHI 5000 Versaprobe, Ulvac-PHI) with an Al-Kα source (1486.6 eV) was performed, where binding energies were calibrated using the C 1$s$ peak from adventitious carbon at 284.6 eV to compensate for the surface charging. To probe the chemical state of the electrode surface, X-ray absorption spectroscopy (XAS) was carried out at the 10D XAS KIST beamline of Pohang Light Source (PLS)-II. To characterize the surface functional moieties on the electrode, attenuated total reflectance-Fourier transform infrared (ATR-FTIR; Alpha II, Bruker) spectroscopy was carried out in the argon-filled glove box. For the preparation of specimens for the above analyses, electrode samples were handled entirely in an Ar-filled glovebox at 25 ± 0.5 °C. For cycled cells, electrode samples were disassembled inside the glovebox, and the harvested electrodes were rinsed thoroughly with the corresponding anhydrous electrolyte solvent used in each cell. While maintaining the

same inert atmosphere, the specimens were then loaded into an air-tight analysis vessel, immediately sealed, and transported to the spectrometer. The gas component from the reaction of TMP treated Mg with deionized water was analyzed by gas chromatography-flame ionization detection (GC-FID; 6000 series, Young Lin). To examine the crystal structure of the materials, X-ray diffraction (XRD; MiniFlex II, Rigaku) patterns were collected using Cu K$\alpha$ radiation ($\lambda = 1.542$ Å) at a scan rate of 2° min$^{-1}$ with a step size 0.02° in the diffraction angle ($2\theta$) range of 10–80°. $^1$H Nuclear magnetic resonance (NMR; Agilent) spectra were recorded on an Ascend 400 spectrometer (600 MHz) using dimethyl sulfoxide (DMSO)-d$_6$ containing 0.03 % (v/v) tetramethylsilane (TMS) (Sigma–Aldrich, 99.9 at.% D) as solvent at room temperature. The moisture content in the electrolyte was measured using a Karl-Fisher titration tools (899 Coulometer, Metrohm). Mg content of the solution containing suspended or dissolved MgO (Sigma-Aldrich, 97%) or Mg(OH)$_2$ (Sigma-Aldrich, 95%) was analyzed by inductive coupled plasma-optical emission spectrometer (ICP-OES) using an iCAP 6500 Duo (Thermo Fisher Scientific).

## Computational details

All first-principles calculations were carried out using the Vienna Ab initio Simulation Package (VASP) with spin-polarized DFT[62]. The Projector Augmented Wave (PAW) approach was used to handle the interactions between electrons and ions[63]. The exchange-correlation energy of the electrons was computed using the Perdew-Burke-Ernzerhof (PBE) functional, a form of the Generalized Gradient Approximation (GGA)[64]. The convergence criterion for ionic relaxation was set to 0.05 eV Å$^{-1}$ based on Hellmann-Feynman forces[65], and electronic relaxation was considered complete when the energy change was less than $1 \times 10^{-4}$ eV. A plane-wave cut-off energy of 400eV was applied to all calculations. To sample the first Brillouin zone in reciprocal space, Monkhorst-Pack grids of (10×10×10) k-points for bulk and (4×4×1) k-points for the slab were used[66]. To compute the reactions at the magnesium electrode surface, a 4×4 Mg(0001) slab with 4 atomic layers was used. The bottom 2 layers were fixed to maintain the bulk lattice constant, while the top 2 layers were allowed to relax. To eliminate interactions between the top and bottom layers due to periodic boundary conditions, a vacuum region equivalent to 9 layers was added along the z-axis.

## Data availability

The data that support the findings of this study are available in the Supplementary Information and Supplementary Data files published alongside this article. Source data are provided with this paper and are available via the Zenodo repository at https://doi.org/10.5281/zenodo.18357206.

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

## Acknowledgements

The authors gratefully thank Dr. Ha, J.-M., of the Korea Institute of Science and Technology for his unstinting support regarding GC-FID measurements. This work was financially supported by the KIST institutional program (2E33942), the National Research Foundation of Korea (RS-2024-00352559, RS-2025-25441257), and the National Research Council of Science & Technology (NST) grant by the Korea government (MSIT) (No. GTL24011-000).

## Author contributions

S. H. Oh conceived the idea and designed the experiments. W. J. No, J. Han, J. Hwang, S. Riedel, and Z. Zhao-Karger performed material synthesis, electrochemical, and structural characterization. H. K. Park and H. C. Ham carried out DFT calculations. M. Jeong, J. Y. Kim, K. Y. Lee, M. Lee, H. D. Yoo, and S.-Y. Lee helped with the battery test and its

analysis. All authors contributed to the data analysis and discussion. S. H. Oh, T. Yim, and H. C. Ham wrote the manuscript.

## Competing interests

The authors declare no competing interests.
