## [Transparent Peer Review file · Nature Communications]

Moisture-tolerant Mg-metal electrodes for practical fabrication of rechargeable Mg batteries

Corresponding Author: Dr Si Hyoung Oh

Version 0:

Reviewer comments:

Reviewer #1

(Remarks to the Author)

The manuscript presents a novel approach to addressing the critical challenge of moisture sensitivity in rechargeable Mg batteries by developing a moisture-tolerant Mg electrode capable of efficient Mg plating-stripping in highly moist electrolytes. The proposed strategy, involving the formation of a sacrificial protection layer via immersion in trimethyl phosphate (TMP), holds promise for reducing manufacturing costs and facilitating the commercialization of Mg batteries.

While the research highlights a compelling and impactful solution to an important barrier in Mg battery technology, there are significant areas that require further attention. The current submission would benefit from a major revision. The following comments and suggestions should be addressed:

- According to the proposed mechanism, the moisture tolerance of the TMP-treated Mg electrode arises from the reaction between Me₂M and H₂O, resulting in the formation of Mg(OH)₂ and methane. Given the high moisture content in some of the studied electrolytes (up to 6500 ppm), this would lead to a significant accumulation of Mg(OH)₂ inside the cell. Since Mg(OH)₂ is a known component of Mg electrode passivation and typically acts as an insulating/ion-blocking layer, how does its formation affect the performance of the cells in this case?

- It is noted that the x-axis (time) does not start at 0 h for some of the plots (Supplementary Fig 6, 7, 10, 12, 13). Is this an error, or was there an OCV period before the GCPL measurements? Is an OCV period or cell resting time necessary to complete the water scavenging process before the measurement is launched? If so, how long is this period, and does it depend on the moisture content?

An additional experiment could be performed where the moist content in the electrolyte before and after (different times of) exposure to the TMP-treated Mg electrode would be measured (by Karl Fisher titration). That would also directly confirm the scavenging effect of the TMP-treated Mg electrode.

Related to this, the Discussion suggests that the moisture scavenging process occurs “during the battery manufacturing process.” However, based on the proposed mechanism, it would seem that scavenging occurs after cell assembly when the electrolyte is in contact with the treated Mg electrode. Could you clarify this point?

- In experiments where cell assembly was done in a dry room, where some water in the atmosphere is still expected, for how long was the pretreated Mg electrode exposed to the atmosphere before cell assembly? It would be interesting to do the cell assembly after different times to gain information on the stability of the formed TMP-based layer in dry room conditions.

- Do you have an estimate of the thickness of the TMP-formed layer? Based on the XPS results, it appears to be at least 10 nm, however, since it was detected by spectroscopy, it is likely to be thicker. In Supplementary Fig. 1B, the morphology seems to exhibit a pitting profile rather than the formation of a smooth, uniform layer.

- Supplementary Fig. 2 suggests that 15 min is the optimal dipping time for Mg electrode treatments, as prolonged dipping causes the metallic luster to wear off. Including an additional photo of the electrode after extended dipping (for example, 20 min) would help confirm this observation. Conversely, the description of Supplementary Fig. 4 indicates that for non-scraped Mg foil, a one-hour dipping is required. What is the reason for this? The non-scraped Mg foil is likely covered with a native passivation layer - does TMP react with it? Another minor comment to Supplementary Fig 4.: parts B and C look very similar. Adding a legend or annotations directly to the figure would make it clearer for readers to distinguish between the two plots

without reading the rather lengthy description in the caption.

- In the final part of the manuscript, TMP-treated Mg electrodes were tested in a full-cell configuration with various cathode materials in both dry and moist electrolytes. For the NVO cathodes, the difference in moisture content between the dry and moist electrolytes was relatively small (20 ppm vs. 60 ppm, respectively). Have you considered testing with electrolytes containing higher moisture content, as was done for Mo₆S₈ and ACC, to allow for a more consistent comparison?
- It would be interesting to explore the compatibility and the performance of the TMP-treated Mg electrode also with state-of-the-art electrolytes with weakly coordinated anions, like Mg alkoxyborates or alkoxyaluminates.
- The introduction could benefit from a more comprehensive review of the topic to provide readers with a broader perspective.

While the manuscript effectively highlights the use of TMP for tailoring the Mg metal surface, it is worth noting that TMP has been previously reported in the literature, primarily as an additive or co-solvent for modifying the solvation structure and influencing interfacial chemistry (for example, *Angew. Chem. Int. Ed.* 2022, 61, e202205187, and *ACS Energy Lett.* 2025, 10, 1, 552–561). Including this context would strengthen the discussion of TMP's relevance in the field.

Additionally, the discussion of moisture-durable electrolytes focuses on earlier generations, such as organohaloaluminates and Mg(TFSI)₂. However, more recent advancements, including Mg alkoxyborates and alkoxyaluminates, could also be highlighted. For example, Mg[Al(hfip)₄]₂ in diglyme electrolyte enabled efficient Mg plating/stripping with up to 1000 ppm of water without a significant increase in overpotential (*ACS Appl. Mater. Interfaces* 2022, 14, 23, 26766–26774).

While moisture tolerance remains a critical challenge for the commercialization of rechargeable Mg batteries, moisture scavenging in electrolytes is a widely employed strategy to address this issue. Organometallic reagents (e.g., Bu₂Mg, Al₃Mg, Mg(BH₄)₂, Mg powder) are frequently added to electrolytes to remove moisture and other impurities (e.g., *J. Electrochem. Soc.* 2015, 162, A7118; *Adv. Energy Mater.* 2024, 14, 2401587; *ACS Energy Lett.* 2017, 2, 1197–1202; *ACS Appl. Mater. Interfaces* 2021, 13, 33123–33132; *J. Mater. Chem. A* 2024, 12, 3386–3397).

Incorporating these additional examples would provide readers with a more contextualized and up-to-date overview of the field.

- The structure of the manuscript could be improved to ensure that conclusions are presented only after supporting results have been introduced and discussed. For instance, in the section “Preparation of moisture-tolerant Mg electrodes via TMP treatment,” the following text appears:

“This simple treatment provided the Mg electrode with highly activated surface states intimately covered with a protective layer abundant in organomagnesium and organophosphate moieties, which allowed the electrode to endure highly moist electrolytes containing a moisture content comparable to where undried salts were dissolved in the solvent (~1000 ppm H₂O). This suggests that the protected electrode can be applied to the cell manufacturing process in the absence of an atmosphere-controlling facility such as a glove box. This can enable fast and cost-effective battery production.”

At this stage in the manuscript, the moieties have not yet been identified, nor has the water tolerance been demonstrated. These statements reflect the general conclusions of the study and would be better suited to the discussion section rather than under this subheading.

Additionally, the beforementioned paragraph transitions to discussing Mg plating-stripping behavior, which would be more logically placed in the subsequent section, “Mg plating-stripping performance ...”. Reorganizing these sections would help maintain a clearer structure and improve the logical flow of the manuscript.

- Some of the experimental details are missing in the Methods section, for example, the amount of the electrolyte used for cell assembly and the number of separators. The amount of the electrolyte is especially important in terms of evaluating the overall water content.
- Supplementary Fig. 5, 7, and 10 could use the same legend (schematics of (treated) Mg electrode, H₂O molecules) as other plots so that the reader can easily recognize the tested systems.
- Solvent abbreviations should be unified. Diglyme is referred to as G2 in the manuscript but DEGDME in the Supplementary Information.

Reviewer #2

(Remarks to the Author)

Reviewer #3

(Remarks to the Author)

In the attachment.

Reviewer #4

(Remarks to the Author)

In this work the authors describe a chemical surface passivation approach using trimethyl phosphate (TMP) for stabilizing Mg metal electrodes to environmental processing and electrolytes with significant water content. Although this work is potentially of interest, TMP has been previously reported as an additive with beneficial properties for Mg batteries (Angew. Chem. Int. Ed. 2023, 62, e202304411). Therefore, more experimental data are needed about the specific mechanism by which this surface passivation strategy succeeds in order to more clearly define how their work differs from previous literature work in this system. Specific comments provided below:

1. How much electrolyte was used to fill the cells used for testing? The surface film imaged in Figure S1 seems far too thin to effectively scavenge 1000s ppm of H₂O from the electrolyte, and quantitative analysis is necessary to explain how much Me₂Mg needs to form to accomplish this task. Analysis of the electrolyte water content after exposure to treated Mg surfaces is also needed to validate this takes place. For example, 6500 ppm H₂O corresponds to ~0.4 M, or approximately half of the total Mg²⁺ present in halide-containing electrolytes and almost the same as in halide-free electrolytes. This seems like far too much water to be accounted for just by scavenging by a surface film.

2. Related to the point above – how does enough Me₂Mg survive exposure to air to still provide any benefit? This Grignard species should be fully reacted almost immediately when exposed to ambient conditions. Characterization after air exposure is needed to understand the surface chemistry of the treated Mg electrodes and confirm the same protection mechanism is active.

3. The primary reaction mechanism proposed with Me₂Mg results in Mg(OH)₂ formation. This is insoluble in ethers, and so it should still passivate the Mg surface. How then is Me₂Mg protecting the surface without simultaneously passivating it?

4. How was the Mg dimethyl phosphate (DMP) prepared? It is not described in the methods and this is important to help validate the proposed formation of Me₂Mg. Extrinsic reaction of Mg(DMP)₂ prior to water exposure could also explain the lack of methane evolution observed in Figure 3b. A more direct demonstration of Me₂Mg formation would be preferable, either by additional spectroscopic analysis or other methodologies, as its presence is central to the proposed mechanism of protection.

5. How do the authors rule out hydrogen evolution/oxidation as the primary electrochemical couple in two electrode cells with hundreds to thousands of ppm H₂O? It is thermodynamically preferred to Mg deposition/stripping and water is present at high enough concentrations to significantly contribute to the measured current. Additional electrochemical experiments are needed, ideally with a three-electrode setup to validate the potentials at which the electrodes are operating, in order to confirm that the observed electrochemistry is actually related to Mg plating/stripping.

Version 1:

Reviewer comments:

Reviewer #1

(Remarks to the Author)

The authors properly addressed all my comments/questions and the manuscript can now be considered for publication.

Reviewer #2

(Remarks to the Author)

Reviewer #3

(Remarks to the Author)

The authors have adequately addressed most of my concerns and manuscript should be suitable for publication after minor revision.

Nevertheless, I would like to authors to comment on the increased overpotential of MgBhfp-based electrolytes (E4) with increasing water content, as well as in E3. Does this mean that there is still a beneficial effect of chloride species in wet electrolytes? Where they able to identify chloride-based species in analysis of their cycled electrodes?

Reviewer #4

(Remarks to the Author)

The authors have satisfactorily addressed several of my initial concerns; however, two key issues must still be addressed before this manuscript can be considered further:

1. The measurements of H₂O and Mg content before and after electrolyte exposure to treated Mg foils are informative, but they do not support a key conclusion of the manuscript. Specifically, the authors hypothesize that Mg(OH)₂ formed from reaction of Me₂Mg with H₂O remains suspended in the electrolyte, but that cannot be the case. The removal of 2000-3000 ppm of H₂O by Me₂Mg should quantitatively result in the formation of 1000-1500 ppm of Mg(OH)₂. However, only 65 ppm Mg is observed in the electrolyte after reaction. This is two orders of magnitude less than would be expected. Where does the Mg(OH)₂ go if it's not on the Mg metal? As I mentioned in my original review, understanding of the protection mechanism is key to differentiating this approach from other work on this system, and these results leave significant ambiguity as to what is actually happening.

2. Three electrode measurements of Mg plating/stripping in water-containing electrolytes are required to validate the proposed mechanism and confirm that Mg plating/stripping is the primary electrochemical process being driven in these electrolytes after exposure to treated metal. As the authors note, there are other species introduced into the electrolyte by contact with the treated electrodes, and there is still significant residual water that remains after reaction (>1000 ppm). Any or all of these species can participate in electrochemical side reactions that will obscure inefficiency in a two electrode measurement, even in an asymmetric cell configuration. Two electrode galvanostatic measurements cannot convincingly rule out side reactions. The authors' claims of practical difficulty in performing three electrode measurements (e.g., water inclusion from the reference electrode) are inconsistent with an extensive body of literature on non-aqueous Mg electrochemistry. Three electrode measurements are quite possible, and they have been demonstrated in systems with Grignard reagents (e.g., 10.1038/35037553) and where water was added intentionally (e.g., 10.1021/acs.chemmater.6b03227). This point again speaks to the need to better define the protection mechanism by ruling out the possibility of side reactions that contribute to the electrochemistry being probed.

Other questions:

1. The authors mention in their response that Me₂Mg would be expected to react with O₂, but that performing work in a dry room prevents this reactivity. The amount of O₂ in the atmosphere of a dry room is not significantly different than in a regular laboratory space. Please clarify this point – if Me₂Mg reacts with O₂, how does it survive the dry room environment? Does some other species form that still results in the release of CH₄ when exposed to H₂O?

2. Beyond the appearance of the peak at 500 cm⁻¹ in dry solvents exposed to treated Mg electrodes, significant changes to both the THF and DME vibrational modes are also observed in the FTIR spectra in Figure S18. What is the origin of these changes (e.g., Me₂Mg coordination with the solvents, decomposition product formation, some other effect)? Additional discussion is needed to explain these changes. A more convincing experiment would be to add water to the dry solvents after exposure to treated Mg to see if the spectra return to what is observed for the unexposed solvents.

Version 2:

Reviewer comments:

Reviewer #4

(Remarks to the Author)

The authors have appropriately addressed my concerns, and I recommend that the manuscript be published after one minor revision. I strongly suggest that the authors include the additional FTIR data provided as part of their response to my comments in the revised SI, as it provides important additional information about the origin and evolution of the DMP content in the electrolyte as a function of water content.

Dear Editor:

Please find an attached revised manuscript we submit for possible publication in *Nature Communications*, titled "Moisture-tolerant Mg electrodes for practical rechargeable Mg batteries".

We revised our manuscript based on the reviewers' comments and submit it according to the previous editorial decision. We hope that we have addressed all the concerns properly which reviewers pointed out. Below we attached our response to the reviewers' comments. The corrected words and sentences in the main text are highlighted in yellow for reviewers' convenience.

Truly yours,

Sincerely,

Si Hyung Oh, PhD

Principal investigator and professor

Energy Storage Research Center, Korea Institute of Science Technology,

5, Hwarang-ro 14-gil, Seongbuk-gu, Seoul 02792, Republic of Korea,

Email- sho74@kist.re.kr

Table of Contents

1. Response to Reviewer #1's comments.....	3
2. Response to Reviewer #2's comments.....	27
3. Response to Reviewer #3's comments.....	28
4. Response to Reviewer #4's comments.....	43

Response to Reviewer #1's comments:

The manuscript presents a novel approach to addressing the critical challenge of moisture sensitivity in rechargeable Mg batteries by developing a moisture-tolerant Mg electrode capable of efficient Mg plating-stripping in highly moist electrolytes. The proposed strategy, involving the formation of a sacrificial protection layer via immersion in trimethyl phosphate (TMP), holds promise for reducing manufacturing costs and facilitating the commercialization of Mg batteries. While the research highlights a compelling and impactful solution to an important barrier in Mg battery technology, there are significant areas that require further attention. The current submission would benefit from a major revision. The following comments and suggestions should be addressed:

⇒ We are grateful to the reviewer for the positive assessment and recognition of the value of this study. We carried out additional experiments and revised our manuscript according to reviewer's insightful and constructive comments.

(1) According to the proposed mechanism, the moisture tolerance of the TMP-treated Mg electrode arises from the reaction between Me_2Mg and H_2O , resulting in the formation of $\text{Mg}(\text{OH})_2$ and methane. Given the high moisture content in some of the studied electrolytes (up to 6500 ppm), this would lead to a significant accumulation of $\text{Mg}(\text{OH})_2$ inside the cell. Since $\text{Mg}(\text{OH})_2$ is a known component of Mg electrode passivation and typically acts as an insulating/ion-blocking layer, how does its formation affect the performance of the cells in this case?

⇒ We are grateful to the reviewer for the insightful comments. We totally agree with the reviewer's opinion that once even thin layer of **compact** $\text{Mg}(\text{OH})_2$ were formed on Mg surface, it could passivate the Mg surface to block further ion transport.

⇒ Firstly, we would like to emphasize that Grignard reagent-like species, Me_2Mg is highly soluble in ethereal solution including G1, and G2 employed in this work. Therefore, it is highly likely that as soon as electrolyte is placed on TMP-treated Mg, Me_2Mg will quickly permeate into the electrolyte to scavenge H_2O molecules instantly, generating

Mg(OH)₂ particles and methane gas in the bulk electrolyte, not on the Mg surface. We believe that these Mg(OH)₂ particles will remain suspended in the electrolyte or be attached to the separators with hydrophilic moieties. Therefore, Mg(OH)₂ particles will not form any compact passivation layer on Mg metal

- ⇒ We dipped TMP-treated Mg metal and scraped Mg metal into the moist G1 solvent (3135 ppm H₂O) for 1 day, respectively. Then the solvent was analyzed to examine any Mg species suspended in the solvent. The result is now shown in Supplementary Table 1. We found 1.33 ppm for the solvent with scraped Mg metal and 65.1 ppm for the solvent with TMP-treated Mg, respectively. This indicates that Mg(OH)₂ was created, being suspended in the solvent from the reaction between moisture and Me₂Mg.
- ⇒ It is also noteworthy that the amount of Me₂Mg in the protective layer is regarded to be high enough to convert thousands ppm of water completely to Mg(OH)₂.
- ⇒ The fact that reversible Mg plating-stripping occurred on TMP-treated Mg or SS working electrode is self-explanatory that no compact Mg(OH)₂ layer exists on Mg metal surface.

Supplementary Table 1. Mg content suspended in the solvent measured by ICP-OES after 5 pieces of TMP-treated or scraped Mg metal discs were immersed in 5 mL of moist G1 with 3135 ppm H₂O for 1 day.

Mg metal in moist G1 (3135 ppm H ₂ O)	Mg content / ppm
Scraped Mg	1.33
TMP-treated Mg	65.1

(2) It is noted that the x-axis (time) does not start at 0 h for some of the plots (Supplementary Fig 6, 7, 10, 12, 13). Is this an error, or was there an OCV period before the GCPL measurements? Is an OCV period or cell resting time necessary to complete the water scavenging process before the measurement is launched? If so, how long is this period, and does it depend on the moisture content?

- ⇒ We are grateful to the reviewer for the careful observation and related comments. In those figures, the x-axis did not start at zero indeed, and it was not an error. There was OCV period of 3 hours before plating-stripping measurements. We did that intentionally to give some reaction time for moisture-scavengers to remove the moisture content from the electrolyte.
- ⇒ To avoid further confusion, we adjusted the starting time of Mg plating-stripping begins at zero in every case, unless mentioned otherwise (Fig. 2c, 2e, 2h, Supplementary Fig. 4 - 13). We also added the following sentence in the experimental. “The cells were put to a resting period of 3 hours before the current was applied unless specified otherwise.”

(100 μL) with various moisture content were applied to the cells with one piece of glass fiber separator to evaluate the moisture tolerance of TMP-treated Mg electrodes. The cells were put to a resting period of 3 h before the current was applied unless specified otherwise. Mg plating-stripping was carried out with 0.1 mA cm^{-2} , and 0.1 mAh cm^{-2} condition,

- ⇒ To address this point out more clearly, we carried out additional experiments. We investigated the reversibility of Mg plating-stripping without those resting periods of 3 hours in various electrolyte content in the electrolyte (300, 997, 3112 ppm). Now it is included in Fig. R-1 and Supplementary Fig. 22.

Fig. R-1. Effect of rest time on Mg plating-stripping performance of TMP-treated Mg electrodes. Comparison of Mg plating-stripping cycling of symmetric cells with TMP-treated Mg electrodes in *E1* electrolyte containing $\sim 1000 \text{ ppm H}_2\text{O}$ depending on the rest time after cell assembly without rest time (red)

or after 3 h of resting (black). The similar performance indicates that moisture scavenging is completed short after the cell assembly.

⇒ In every case, the cells started to operate normally without much difficulty. Considering that the time necessary for cell assembling and installing in the multi-channel tester takes about 10 minutes, this result indicates that the scavenging process by Me_2Mg is very effective and the time for scavenging moisture in the electrolyte is very short.

Supplementary Fig. 22. Mg plating-stripping behavior of symmetric cells using TMP-treated Mg electrodes in *EI* containing (A) 300 ppm, (B) 997 ppm, and (C) 3112 ppm of water. All measurements were conducted without resting period after cell assembly. Stable Mg plating-stripping performance under this condition (no resting time) indicated that moisture-scavenging by Me_2Mg in the moist electrolyte was very fast and highly effective.

(3) An additional experiment could be performed where the moist content in the electrolyte before and after (different times of) exposure to the TMP-treated Mg electrode would be measured (by Karl Fisher titration). That would also directly confirm the scavenging effect of the TMP-treated Mg electrode.

- ⇒ We are grateful to the reviewer for the insightful comments and suggesting important additional experiment. As requested, we carried out the experiments for confirming the scavenging effect of the TMP-treated Mg. For this purpose, we prepared two 0.5 mL of highly moist electrolytes containing 3700 ppm and 6600 ppm H₂O, respectively. 0.5 mL is the minimum volume with which moisture content was reliably evaluated with Karl-Fisher titration tools. In each of these electrolytes, five pieces of TMP-treated Mg disc were inserted for three hours (corresponding to 100 μL of electrolyte per TMP-treated Mg).
- ⇒ As shown in Supplementary Table 2, the scavenged H₂O amounted to 2100 ~ 2800 ppm. This indicates that TMP-treated Mg hold the capability of scavenging at least 2100 ~ 3000 ppm of moistures from the electrolyte in 3 hours.
- ⇒ This is a strong evidence that TMP-treated Mg actually scavenges moisture from the electrolyte, lowering down moisture-level of the electrolyte system enough to enable reversible Mg plating-stripping.

⇒

Supplementary Table 2. Residual H₂O content in 0.5 mL of electrolyte measured by Karl-Fisher titration tools after immersing 5 pieces of TMP-treated Mg metal discs for 3 h.

Initial H ₂ O content ↓ / ppm	Final H ₂ O content ↓ / ppm	Amount of H ₂ O scavenged ↓ / ppm
3734.9	1622.4	2112.5
6606.4	3774.4	2832.0

(4) Related to this, the Discussion suggests that the moisture scavenging process occurs “during the battery manufacturing process.” However, based on the proposed mechanism,

it would seem that scavenging occurs after cell assembly when the electrolyte is in contact with the treated Mg electrode. Could you clarify this point?

- ⇒ We are grateful to the reviewer for the insightful comments on this matter. The reviewer pointed out that moisture scavenging process might occur only after the cell assembly when the electrolyte was actually in contact with the treated Mg electrode based on the experiments that had been carried out in this work.
- ⇒ What we intended to say in Discussion is that TMP-treated Mg can tolerate the moisture contamination from the air or the moist electrolyte during the cell manufacturing process.
- ⇒ To clarify this, we carried out the experiments for evaluating the reversibility of Mg plating-stripping after TMP-treated Mg was exposed to dry-room air in different period of time (5 min, 30 min, 1 h, 3 h). After set period of time, the cell was assembled within 1 min in a dry room facility. The electrolyte containing 997 ppm of H₂O was applied to evaluate the moisture-tolerance after a span of exposure time to air. As shown in Fig. 5a, the reversibility was relatively well-maintained up to 1 hour of exposure time. They lost moisture-tolerance clearly after 3 hours' exposure in the dry room atmosphere as overpotential for Mg plating-stripping increased sharply. This indicated that scavenging agent (Me₂Mg) consistently eliminated moisture and oxygen content during exposure to air.
- ⇒ These results indicate that TMP-treated Mg can cope with the moisture contamination from both the air or the moist electrolyte at the same time during the cell manufacturing process. Therefore, we believe that in this study, moisture–scavenging occurred throughout the battery manufacturing process.

Fig. 5. Characterization of TMP-treated Mg electrodes pre-exposed to dry room air for different time period.

(a) Mg plating-stripping behavior of symmetrical cells in *E1* (997 ppm H₂O) composed of a pair of TMP-treated Mg electrodes, which was pre-exposed to dry room air for 5 min., 30 min., 1 h, and 3 h. (b) GC-FID spectra for the gases that evolved after the TMP-treated Mg electrode, which were pre-exposed to dry room air for 0 min., 5 min., 30 min., and 3 h, respectively, were dipped in moist electrolyte (*E1*, 997 ppm H₂O).

(5) In experiments where cell assembly was done in a dry room, where some water in the atmosphere is still expected, for how long was the pretreated Mg electrode exposed to the atmosphere before cell assembly? It would be interesting to do the cell assembly after different times to gain information on the stability of the formed TMP-based layer in dry room conditions.

⇒ We are grateful to the reviewer for raising this important and interesting issue. As requested, we carried out the experiments for evaluating the reversibility of Mg plating-stripping after TMP-treated Mg was exposed to dry room air in different period of time (5 min, 30 min, 1 h, 3 h). After set period of time, the cell was assembled within 1 min in a dry room facility. The electrolyte containing 997 ppm of H₂O was applied to evaluate the moisture-tolerance after a span of exposure time to air. As shown in Fig. 5a, the reversibility was relatively well-maintained up to 1 hour of exposure time. They lost moisture-tolerance apparently after 3 hours' exposure in the dry room atmosphere as overpotential for Mg plating-stripping increases sharply. This shows that scavenging

agent (Me_2Mg) consistently eliminates moisture and oxygen content during exposure to air.

Fig. 5. Characterization of TMP-treated Mg electrodes pre-exposed to dry room air for different time period.

(a) Mg plating-stripping behavior of symmetrical cells in *E1* (997 ppm H_2O) composed of a pair of TMP-treated Mg electrodes, which was pre-exposed to dry room air for 5 min., 30 min., 1 h, and 3 h. (b) GC-FID spectra for the gases that evolved after the TMP-treated Mg electrode, which were pre-exposed to dry room air for 0 min., 5 min., 30 min., and 3 h, respectively, were dipped in moist electrolyte (*E1*, 997 ppm H_2O).

⇒ We added Mg plating-stripping feature for 5 min ~1 h exposure to dry-room air to Fig. 5a, and corresponding discussion in “Discussion” of the main text to emphasize the actual applicability of TMP-treated Mg metal anode to dry room manufacturing process of Mg-ion batteries.

Discussion

To enable practical dry-room manufacturing of RMBs using the strategy developed in this work, it is essential to secure a sufficient protection time window for cell assembly, given the high reactivity of Me_2Mg with ambient moisture and oxygen. To this end, the reversibility of Mg plating-stripping was evaluated in moist electrolyte (997 ppm H_2O) with TMP-treated Mg that had been exposed to dry room air for various durations. As shown in Fig. 5a, the reversibility was relatively well maintained up to 1 h of air exposure. However, after 3 h, moisture-tolerance was clearly lost, as evidenced by a sharp increase in the overpotential during Mg plating-stripping. GC-FID analysis of the gases that evolved from the reaction between the moist electrolyte and TMP-treated Mg, which was pre-exposed to dry-room air for different durations, revealed that a substantial amount of CH_4 was still generated after 30 min. of air exposure (Fig. 5b, Supplementary Fig. 31). These results indicate that the scavenging agent, Me_2Mg , not only removed moisture from the electrolyte, but also continuously eliminated moisture and oxygen from the environment during the entire manufacturing process. This highlights the potential of our strategy to significantly improve the feasibility of dry-room manufacturing of RMBs.

(6) Do you have an estimate of the thickness of the TMP-formed layer? Based on the XPS results, it appears to be at least 10 nm, however, since it was detected by spectroscopy, it is likely to be thicker. In Supplementary Fig. 1B, the morphology seems to exhibit a pitting profile rather than the formation of a smooth, uniform layer.

- ⇒ We are grateful to the reviewer for the careful observation and associated comments. We agree with the reviewer's opinion that there formed about 10 μm of protective layer on Mg metal from the 15 minute's treatment of Mg in TMP, which can be confirmed from the SEM cross-section images of Supplementary Fig. 1B. It is also true that the reaction between TMP and Mg metal led to pitting-like activation surface
- ⇒ Actually, we have also noticed this issue, and continued further research on this subject, focusing on creating uniform protective surface on Mg surface. By treating Mg metal in BiCl_3 -dissolved TMP solution, we could obtain a smooth and glossy surface of thin multi-layered composite protective film consisting of Bi (or Bi_2Mg_3) metal layer followed by $\text{Mg}(\text{DMP})_2$ (+ Me_2Mg) layer. As shown in Fig. R-2A~C, Mg surface is covered with thin Bi layer followed by normal $\text{Mg}(\text{DMP})_2$ in the figure. We also observed that this

method is useful to apply to large-area electrode, while maintaining a similar moisture-tolerance capability of TMP-treated Mg. We also found that other metals like Zn has some effect in homogenizing the surface.

⇒ We have a plan to report this result separately stressing on manufacturing a large-sized pouch cell in a dry room facility. By this reason, we would like to focus on reporting the mechanism and potential as an effective technique that can realize the dry room manufacturing of Mg-ion batteries in this paper.

Fig. R-2. (A) Digital photograph of a BiCl_3 -TMP-treated Mg electrode, (B) Top-view SEM image of the electrode surface with EDS elemental mapping results, (C) Cross-sectional SEM image of the electrode and corresponding EDS elemental mapping.

(7) Supplementary Fig. 2 suggests that 15 min is the optimal dipping time for Mg electrode treatments, as prolonged dipping causes the metallic luster to wear off. Including an additional photo of the electrode after extended dipping (for example, 20 min) would help confirm this observation.

⇒ We are grateful to the reviewer for this comment. We added additional photos for TMP-treated Mg (20 min and 30 min) in Supplementary Fig. 2. As seen in these photos, extended treatment in TMP led to loss of metallic luster, uneven morphology and excessive wear on the surface.

Supplementary Fig. 2. Photographs of Mg metal foil disks before and after TMP treatment (after 1 seconds and after up to 30 minutes). The initial metallic luster began to wear off as reaction time went by. After 1 day of dipping in TMP, Mg metal was completely dissolved.

(8) Conversely, the description of Supplementary Fig. 4 indicates that for non-scraped Mg foil, a one-hour dipping is required. What is the reason for this? The non-scraped Mg foil is likely covered with a native passivation layer - does TMP react with it?

⇒ We are grateful to the reviewer for the insightful comment. As the reviewer also pointed out, we believe that non-scraped Mg foil was likely covered with native passivation layer initially. We believe that it was Mg metal that TMP actually reacted with, and native oxide seemed to impede the reaction between Mg metal and TMP. To clarify this, we examined the surface of the TMP-treated non-scraped Mg metal foil, and the results are shown in Supplementary Fig. 34.

- ⇒ For non-scraped Mg metal, reaction with TMP resulted in the development of reaction pits. The gradual increase in number and size of reaction pits was observed over time. After 1 h of reaction time, the pit size grew to $\sim 100 \mu\text{m}$. But, after 3 h of reaction time, the reaction seemed to too excessive as many bored-through pits ($\sim 500 \mu\text{m}$ in diameter) were observed. This indicate that the reaction rate between TMP and Mg is substantially lower without peeling off the native oxide layer.
- ⇒ We also analyzed the solubility of possible constituents of passivation layer, MgO and Mg(OH)₂ in the TMP by placing the extra amount of MgO or Mg(OH)₂ species into TMP. Around 40 ppm of Mg species were detected from the ICP-OES analysis in both cases. This means that solubility of MgO or Mg(OH)₂, is quite low in TMP. Therefore, the reaction between TMP and non-scraped Mg proceeds slowly probably because it takes considerable time for TMP molecules to penetrate into the bare Mg surface along the defect sites on the surface or grain boundary.

Supplementary Fig. 34. SEM images and photographs of non-scraped Mg metal foil disks before and after TMP treatment (15 min, 1 h, and 3 h). For non-scraped Mg metal, reaction with TMP resulted in the development of reaction pits. The gradual increase in number and size of reaction pits was observed over time. After 1 hour of reaction time, the pit size grew up to $\sim 100 \mu\text{m}$. But, after 3 h of reaction time, the reaction seemed to too excessive as many large bored-through pits ($\sim 500 \mu\text{m}$ in diameter) were observed.

Supplementary Table 3. Analysis on the Mg content dissolved in TMP solution after extra amount of MgO or Mg(OH)₂ powder was added to the solution. The concentration was measured by ICP-OES.

Mg species in TMP	Mg content / ppm
Saturated MgO	42
Saturated Mg(OH) ₂	40

(9) Another minor comment to Supplementary Fig 4.: parts B and C look very similar. Adding a legend or annotations directly to the figure would make it clearer for readers to distinguish between the two plots without reading the rather lengthy description in the caption.

⇒ We are grateful to the reviewer for pointing out this matter. We corrected Supplementary Fig. 4: parts B and C as reviewer requested.

Supplementary Fig. 4. Electrochemical characterization of TMP-treated Mg electrodes in moist (314 ppm H₂O) | E2 electrolyte (0.5M Mg(TFSI)₂ + 0.5 M MgCl₂ in G2). Mg plating-stripping behavior with of Mg||SS asymmetric cells composed (A) of scraped untreated Mg electrode, and a stainless steel foil working electrode, (B) of TMP-treated Mg electrode, and a stainless steel foil working electrode, and (C) of TMP-treated Mg(non-scraped) electrode, and a stainless steel foil working electrode. TMP-treated Mg(non-scraped) electrode was prepared by reacting TMP with a Mg foil disc for one hour whose native oxide was not removed beforehand. In contrast to scraped Mg, it took at least one hour to activate non-scraped Mg electrode with TMP, as evidenced by the gradual increase in the number and size of reaction pits over time (Supplementary Fig. 34). The reaction between TMP and non-scraped Mg proceeded slowly, likely because the native oxide layer was insoluble in TMP (Supplementary Table 3) and it took considerable time for TMP molecules to penetrate to the bare Mg surface along the grain boundary or defect sites on the surface.

(10) In the final part of the manuscript, TMP-treated Mg electrodes were tested in a full-cell configuration with various cathode materials in both dry and moist electrolytes. For the NVO cathodes, the difference in moisture content between the dry and moist electrolytes was relatively small (20 ppm vs. 60 ppm, respectively). Have you considered testing with electrolytes containing higher moisture content, as was done for Mo₆S₈ and ACC, to allow for a more consistent comparison?

⇒ We are grateful to the reviewer for this comment. It is true that for NVO, the difference in moisture content of the electrolyte was relatively small compared to Mo₆S₈ and ACC. Actually, we have evaluated the performance of NVO with a higher moisture content (600 ppm). The result was shown now in Fig. R-3 below. In the highly moist electrolyte, the performance of NVO was not as good as the one in the electrolyte with lower moisture contents. We believe that this was caused by incorporation of some water molecules into the interlayers of NVO in the moist electrolyte, which is a common

phenomenon in the aqueous electrolyte (Nat. Energy 1, 16119 (2016)). These water molecules in the interlayer may not be easily scavenged by Me_2Mg . It seemed that these water molecules are decomposed during the charge process judging from the large charging capacity during the early cycles.

⇒ We believe that every cathode material may have its own characteristics towards moisture, so that it could be unfair to apply a uniform moisture standard to all cases. While some materials are corrosion-resistant or even benefited by the presence of water, others may not be. The primary target in this work is to realize moisture-tolerable Mg battery manufacturing process in a dry room facility. The intrinsic moisture stability of NVO may not be the primary subject of this study and we will report further studies related to this issue in our future communications.

Fig. R-3. The discharge-charge profiles of a full cell made of $\text{NH}_4\text{V}_4\text{O}_{10}$ cathode and TMP-treated Mg metal anode in moist **E3** electrolyte (600 ppm H_2O). The cell was assembled in an Ar-filled glovebox. The current rate was 0.1C (1C = 400 mA g^{-1})

(11) It would be interesting to explore the compatibility and the performance of the TMP-treated Mg electrode also with state-of-the-art electrolytes with weakly coordinated anions, like Mg alkoxyborates or alkoxyaluminates.

- ⇒ We are grateful to the reviewer for proposing this interesting experiment involving the applicability of our strategy to the electrolytes with weakly coordinated anions. For this purpose, we carried out cooperative studies with Prof. Zhao-Karger from KIT, Germany, who is one of the earliest developers of these electrolyte systems and provided us with $\text{Mg}[\text{B}(\text{hfip})_4]_2$. Using the salt, we prepared a series of electrolytes, **E4**, 0.3 M $\text{Mg}[\text{B}(\text{hfip})_4]_2$ in G1 having moisture content of 2.0, 1092 and 3482 ppm H_2O , which are representative electrolytes among electrolytes with weakly coordinated anions.
- ⇒ We carried out Mg plating-stripping with a symmetrical configuration using TMP-treated Mg electrodes in **E4** electrolytes. The result is shown in Supplementary Fig. 32 which indicates that TMP-treated Mg also work in **E4** electrolytes, although moderate increase in the overpotential observed in the moist conditions. These results were also confirmed by Zhao-Karger's group independently.
- ⇒ To emphasize the versatility of our strategy to various electrolyte systems, we included these experimental results and associated discussions in "Discussion" of the main text.

Electrolytes based on weakly-coordinated anions, such as ethereal solution of Mg alkoxyborates and alkoxyaluminates, have attracted considerable research interest in recent years.^{31, 38, 39} In particular, $\text{Mg}[\text{Al}(\text{hfip})_4]_2$ in G2 (hfip: 1,1,1,3,3,3-hexafluoroisopropoxy) has demonstrated efficient Mg plating-stripping performance even in the presence of 1000 ppm of water, without a significant increase in overpotential.³⁹ To assess the compatibility of our strategy with such electrolytes, the reversibility of Mg plating-stripping was investigated in **E4** electrolyte: 0.3 M $\text{Mg}[\text{B}(\text{hfip})_4]_2$ in G1, containing moisture levels of 2.0, 1092 and 3482 ppm H_2O , respectively – a system whose moisture tolerance has been relatively unexplored. With TMP-treated Mg electrodes, reversible Mg plating-stripping was still achievable even under highly humid conditions (up to ~3500 ppm H_2O), although a moderate increase in overpotential was observed (Supplementary Fig. 32). These results demonstrate the versatility of our strategy across various electrolyte systems.

- ⇒ In recognition of prof. Zhao-Karger's important contribution, we added prof. Zhao-Karger, and Dr. Sibylle Riedel (who carried out actual experiments) as co-authors of this paper.

Supplementary Fig. 32. Mg plating-stripping performance of symmetric cells with TMP-treated Mg electrodes in Mg[B(hfip)₄]₂-based *E4* electrolyte containing different water contents. (A, C, E) Long-term cycling behavior using *E4* electrolytes containing 2 ppm, 1092 ppm, and 3482 ppm of water, respectively. (B, D, F) Enlarged views of the first 20 cycles from (A), (C), and (E), respectively. The plating-stripping was carried out in 0.1 mA cm⁻², 0.1 mAh cm⁻².

(12) The introduction could benefit from a more comprehensive review of the topic to provide readers with a broader perspective. While the manuscript effectively highlights the use of TMP for tailoring the Mg metal surface, it is worth noting that TMP has been previously reported in the literature, primarily as an additive or co-solvent for modifying the solvation structure and influencing interfacial chemistry (for example, *Angew. Chem. Int.*

Ed. 2022, 61, e202205187, and ACS Energy Lett. 2025, 10, 1, 552–561). Including this context would strengthen the discussion of TMP's relevance in the field.

⇒ We are grateful to the reviewer for this valuable suggestion. We were aware that TMP was previously reported in the literature as an additive or co-solvent (Angew. Chem. Int. Ed. 2022, 61, e202205187, and ACS Energy Lett. 2025, 10, 1, 552–561). We added these references in the Reference section.

56. Zhao W, *et al.* Tailoring Coordination in Conventional Ether-Based Electrolytes for Reversible Magnesium-Metal Anodes. *Angewandte Chemie International Edition* **61**, e202205187 (2022).

57. Zhang M, Zhao W, Liu Y, Zhou M, Pan Z, Yang X. Contact Ion-Pair-Dominated Electrolyte Enabling Inorganic-Rich Solid–Electrolyte Interphase for Long-Cycling Magnesium Metal Anodes. *ACS Energy Letters* **10**, 552-561 (2025).

⇒ As the reviewer pointed out, they primarily focused either on modifying the solvation structure by introducing TMP in the solvation sheath, or influencing interfacial chemistry by constructing ion-transporting inorganic SEI rich in fluorides and phosphides. But they did not discuss the aspect of moisture-tolerance and development of inexpensive, scalable manufacturing process of RMBs. On the contrary, our work was much focused on developing a viable strategy for securing moisture tolerance through simple and convenient surface activation in TMP. Therefore, the role, purpose, an application mode of TMP is very different from the previous works, although their works also has tremendous value of their own.

⇒ Since introduction part of this study mostly covered topics associated with moisture-tolerance, we added brief discussions on the difference and importance of our study compared to the previous studies in the “Discussion” of the main text.

It is noteworthy that TMP has often been reported as an additive or co-solvent of the electrolyte for Mg-ion batteries.^{56, 57} These studies primarily focused on either modifying the solvation structure by incorporating TMP molecules in the solvation sheath, or influencing interfacial chemistry by constructing ion-conductive inorganic SEI layers rich in fluorides and phosphides. However, aspect such as moisture-tolerant manufacturing process for RMBs were not seriously addressed. In contrast, the present work focuses on developing a viable strategy to secure moisture tolerance through a simple and convenient surface activation process in TMP, thereby enabling dry room manufacturing of RMBs.

(13) Additionally, the discussion of moisture-durable electrolytes focuses on earlier generations, such as organohaloaluminates and Mg(TFSI)₂. However, more recent advancements, including Mg alkoxyborates and alkoxyaluminates, could also be highlighted. For example, Mg[Al(hfip)₄]₂ in diglyme electrolyte enabled efficient Mg plating/stripping with up to 1000 ppm of water without a significant increase in overpotential (ACS Appl. Mater. Interfaces 2022, 14, 23, 26766–26774). While moisture tolerance remains a critical challenge for the commercialization of rechargeable Mg batteries, moisture scavenging in electrolytes is a widely employed strategy to address this issue. Organometallic reagents (e.g., Bu₂Mg, Al₃Mg, Mg(BH₄)₂, Mg powder) are frequently added to electrolytes to remove moisture and other impurities (e.g., J. Electrochem. Soc. 2015, 162, A7118; Adv. Energy Mater. 2024, 14, 2401587; ACS Energy Lett. 2017, 2, 1197–1202; ACS Appl. Mater. Interfaces 2021, 13, 33123–33132; J. Mater. Chem. A 2024, 12, 3386–3397). Incorporating these additional examples would provide readers with a more contextualized and up-to-date overview of the field.

⇒ We are grateful to the reviewer for this valuable comment. As the reviewer pointed out, the electrolytes based on the weakly-coordinated anions such as ethereal solution of Mg alkoxyborates and alkoxyaluminates, have been intensive research area in recent years. Particularly, Mg[Al(hfip)₄]₂ in G2 electrolyte enabled efficient Mg plating/stripping with up to 1000 ppm of water without a significant increase in overpotential (ACS Appl. Mater. Interfaces 2022, 14, 23, 26766–26774). We added a short discussion on this subject briefly in the introduction as below to help the reader to comprehend the content of this paper.

plating-stripping could be realized only in the absolute moisture-free environment.³⁷ More recently, alkoxyaluminate salts have emerged as highly moisture-resistant electrolytes, demonstrating stable Mg plating-stripping performance in the presence of up to 1000 ppm of water without a notable increase in overpotential.^{31, 38, 39} While moisture-resistance of the electrolyte system remains a critical challenge for the commercialization of RMBs, moisture scavenging in electrolytes is another widely employed strategy to address this issue. Organometallic reagents (e.g., Bu_2Mg , Al_3Mg , $\text{Mg}(\text{BH}_4)_2$, Mg powder) were often added to electrolytes to remove moisture and other impurities.⁴⁰ ⁴³ In most approaches, however, achieving good moisture-tolerance requires rigorous drying procedure for all battery components as well as a strictly controlled manufacturing environment. These measures inevitably lead to a substantial

- ⇒ We also carried out additional experiment that our strategy can work with ethereal solution of Mg alkoxyborates, moisture tolerance of which was not well reported in the previous literature. We found that TMP-treated Mg electrode, reversible Mg plating-stripping was realized with Mg alkoxyborates with 3000 ppm H_2O . This was already discussed in response to comment 11.
- ⇒ As reviewer also pointed out, the moisture scavenging in the electrolyte is a widely employed strategy to secure moisture tolerance. Organometallic reagents (e.g., Bu_2Mg , AlCl_3/Mg powder, $\text{Mg}(\text{BH}_4)_2$, $\text{Al}(\text{CH}_3)_3$) are frequently added to electrolytes to remove moisture and other impurities (e.g., J. Electrochem. Soc. 2015, 162, A7118; Adv. Energy Mater. 2024, 14, 2401587; ACS Energy Lett. 2017, 2, 1197–1202; ACS Appl. Mater. Interfaces 2021, 13, 33123–33132; J. Mater. Chem. A 2024, 12, 3386–3397). We added related discussion briefly in the main text and inserted these papers to the reference list. We also added a brief discussion on this aspect in the introduction as shown above. In this paper, we developed more convenient and effective strategy of scavenging moisture in the electrolyte to realize dry-room manufacturing process for RMBs.

38. Pavčnik T, Imperl J, Kolar M, Dominko R, Bitenc J. Evaluating the synthesis of Mg [Al(hfip)₄]₂ electrolyte for Mg rechargeable batteries: purity, electrochemical performance and costs. *Journal of Materials Chemistry A* **12**, 3386-3397 (2024).
39. Pavčnik Ta, *et al.* On the practical applications of the magnesium fluorinated alkoxyaluminate electrolyte in Mg battery cells. *ACS applied materials & interfaces* **14**, 26766-26774 (2022).
40. Luo J, He S, Liu TL. Tertiary Mg/MgCl₂/AlCl₃ inorganic Mg²⁺ electrolytes with unprecedented electrochemical performance for reversible Mg deposition. *ACS Energy Letters* **2**, 1197-1202 (2017).
41. Shterenberg I, *et al.* Evaluation of (CF₃SO₂)₂N⁻ (TFSI) based electrolyte solutions for Mg batteries. *Journal of The Electrochemical Society* **162**, A7118 (2015).
42. Li Z, *et al.* Establishing a stable anode–electrolyte interface in Mg batteries by electrolyte additive. *ACS applied materials & interfaces* **13**, 33123-33132 (2021).
43. Radi M, *et al.* A Comprehensive Study on the Parameters Affecting Magnesium Plating/Stripping Kinetics in Rechargeable Mg Batteries. *Advanced Energy Materials* **14**, 2401587 (2024).

(14) The structure of the manuscript could be improved to ensure that conclusions are presented only after supporting results have been introduced and discussed. For instance, in the section “Preparation of moisture-tolerant Mg electrodes via TMP treatment,” the following text appears:

“This simple treatment provided the Mg electrode with highly activated surface states intimately covered with a protective layer abundant in organomagnesium and organophosphate moieties, which allowed the electrode to endure highly moist electrolytes containing a moisture content comparable to where undried salts were dissolved in the solvent (~1000 ppm H₂O). This suggests that the protected electrode can be applied to the cell manufacturing process in the absence of an atmosphere-controlling facility such as a glove box. This can enable fast and cost-effective battery production.”

At this stage in the manuscript, the moieties have not yet been identified, nor has the water tolerance been demonstrated. These statements reflect the general conclusions of the study and would be better suited to the discussion section rather than under this subheading.

⇒ We are grateful to the reviewer for this valuable suggestion. We agree with idea that it was too early to place the general conclusion in that place of the manuscript. Therefore, we revised the manuscript so that general conclusions of the study do not appear in that place mentioned.

min at room temperature (Fig. 1c, see **Materials and Methods**, and Supplementary Fig. 1–3 for details). This simple treatment enabled the Mg electrode to endure highly moist electrolytes containing a moisture content comparable to where undried salts were dissolved in the solvent (~1000 ppm H₂O). This suggests that the protected electrode can be applied to the cell manufacturing process in the absence of an atmosphere-controlling facility such as a glove box, enabling fast and cost-effective battery production. Fig. 1d, and 1e compared the Mg plating-stripping behaviors of

(15) Additionally, the beforementioned paragraph transitions to discussing Mg plating-stripping behavior, which would be more logically placed in the subsequent section, “Mg plating-stripping performance ...”. Reorganizing these sections would help maintain a clearer structure and improve the logical flow of the manuscript.

⇒ We are grateful to the reviewer for this valuable suggestion. For this purpose, we combined two sections to make one larger section, because two sections are closely related. The heading for new larger section is “Preparation of moisture-tolerant Mg electrodes via TMP treatment and Mg plating-stripping performance under the moist electrolytes”.

Results

Preparation of moisture-tolerant Mg electrodes via TMP treatment and Mg plating-stripping performance under the moist electrolytes

(16) Some of the experimental details are missing in the Methods section, for example, the amount of the electrolyte used for cell assembly and the number of separators. The amount of the electrolyte is especially important in terms of evaluating the overall water content.

⇒ We are grateful to the reviewer for pointing out this matter. The missing information on the experimental details in the Methods section have been now supplemented.

⇒ The amount of electrolyte applied to each coin cell was set to be 100 μL , and one piece of Whatman glass fiber was placed in each coin cell. We added these details in the Methods section.

was maintained below $-60\text{ }^{\circ}\text{C}$. A glass fiber (GF/F, Whatman) was utilized as a separator. Ether-based electrolytes (100 μL) with various moisture content were applied to the cells with one piece of glass fiber separator to evaluate the moisture tolerance of TMP-treated Mg electrodes. The cells were put to a resting period of 3 h before the current was applied unless specified otherwise. Mg plating-stripping was carried out with 0.1 mA cm^{-2} , and 0.1 mAh cm^{-2} condition,

⇒ We are aware of that the amount of electrolyte is a critical factor in quantifying the total water content in the cell. Accordingly, we evaluated whether Me_2Mg from the TMP-treated Mg can scavenge all the water in the electrolyte quantitatively. The results are displayed in Table R-1.

⇒ From the analysis of cross-sectional image of TMP-treated Mg electrode, the average thickness of Mg metal reacted is around $10\text{ }\mu\text{m}$. The corresponding amount of Mg metal is around $150\text{ }\mu\text{mol}$ per Mg disc of 16 mm diameter. According to the work of P. Becher (J. Am. Chem. Soc. 86 (1964) 1782), 10% of Mg metal reacted was transformed to Me_2Mg . Therefore, the amount of Me_2Mg in the protective film is estimated to be around $15\text{ }\mu\text{mol}$ per Mg disc. As shown in Table R-1, The amount of Me_2Mg necessary for scavenging water content in $100\text{ }\mu\text{L}$ of electrolyte containing $3000\text{ ppm H}_2\text{O}$ is around $10\text{ }\mu\text{mol}$. This is less than the amount of Me_2Mg expected in TMP-treated Mg metal disc. Therefore, TMP-treated Mg electrode has the capability of scavenging at least $100\text{ }\mu\text{L}$ of electrolyte containing $3000\text{ ppm H}_2\text{O}$ (maximum capacity is estimated to be $\sim 4500\text{ ppm}$).

⇒ Meanwhile, the amount of Me_2Mg necessary for eliminating moisture from the electrolyte having $6500\text{ ppm H}_2\text{O}$ was estimated to be $22\text{ }\mu\text{mol}$, which is larger than the Me_2Mg expected in TMP-treated Mg disc. In this case, we believe that along with the scavenging effect of Me_2Mg , $\text{Mg}(\text{DMP})_2$ may act as a physical barrier to protect the Mg surface by adsorbing H_2O on its surface.

Table R-1. Amount of Me_2Mg required to scavenge different levels of water in $100\text{ }\mu\text{L}$ of el

electrolyte. Values were calculated based on the 1:2 molar ratio of Me_2Mg reacting with H_2O .

H_2O in ppm	H_2O in μmol	Required Me_2Mg in μmol
30	0.201	0.101
300	2.012	1.006
1,000	6.706	3.353
3,000	20.12	10.06
6,500	43.59	21.79

(17) Supplementary Fig. 5, 7, and 10 could use the same legend (schematics of (treated) Mg electrode, H_2O molecules) as other plots so that the reader can easily recognize the tested systems.

⇒ We are grateful to the reviewer for this comment. As pointed out, we inserted the same legend (schematics of (treated) Mg electrode, H_2O molecules) in Supplementary Fig. 5, 7, and 10. We appreciate to the reviewer again for the careful reading of our manuscript and suggesting valuable improvements.

(18) Solvent abbreviations should be unified. Diglyme is referred to as G2 in the manuscript but DEGDME in the Supplementary Information.

⇒ We are grateful to the reviewer for this comment. As pointed out, we used G2 instead of DEGDME throughout the paper.

Response to Reviewer #2's comments:

⇒ We are deeply grateful to the reviewer for the valuable feedback and suggestions, which will help improve the quality of this paper.

Response to Reviewer #3's comments:

The authors research important point, namely, the improvement of the compatibility of Mg metal anodes by surface treatment with alkyl phosphate additives. They showcase that use of TMP surface treatment can improve Mg metal anode compatibility with dry room atmosphere and wet electrolytes. The topic is timely and addresses an important research question. However, the same additives have already been reported in field of Mg batteries, with the main mechanism presumably being the altered solvation shell of Mg²⁺ ions. 1,2 I think the authors should aim and clarify difference between their and literature reports to better assign contributions to either bulk electrolyte solvation or Mg metal interphase. I think this should significantly strengthen the impact of the work.

- ⇒ We are grateful to the reviewer for the valuable opinion. We were aware that TMP was previously reported in the literature as an electrolyte additive or co-solvent, for example, *Angew. Chem. Int. Ed.* 2022, 61, e202205187, and *ACS Energy Lett.* 2025, 10, 1, 552–561. While TMP was utilized as an electrolyte additive in those studies, their main roles in their studies were either to modify the solvation structure of Mg²⁺ by introducing TMP molecule in the solvation sheath, or to influence interfacial chemistry by constructing ion-transporting inorganic SEI rich in fluorides and phosphides.
- ⇒ By contrast, in this work, TMP was NOT used as any kind of additive or solvent of the electrolytes. TMP was simply utilized as a surface activating agent for Mg metal anode. No TMP was added to the electrolytes when the electrochemical performance was evaluated in this study. Therefore, there is no intentional alteration in the solvation shell of Mg²⁺, when the electrolytes like **E1**, **E2**, **E3** was applied to TMP-treated Mg electrodes. The purpose of TMP treatment of Mg metal was to construct a moisture-tolerating protective film on the surface of Mg metal that can be applied to the cell assembly to enable dry room manufacturing process of RMBs. The resulting composite surface film consisted of moisture-scavenging Me₂Mg agents dispersed in Mg(DMP)₂ matrix. No TMP molecules exist in the composite protective film on Mg metal when it was applied to cell assembly. As such, while this study is distinct from the previous studies, it can be broadly categorized as being related to controlling Mg metal interphase.

- ⇒ Furthermore, prior studies have not examined the moisture tolerance in-depth and had not intended to develop inexpensive, scalable manufacturing process of RMBs. On the contrary, this study was much focused on developing a viable strategy for securing moisture tolerance through simple and convenient surface activation in TMP. Therefore, the role, purpose, an application mode of TMP is very different from the previous works, although their works also has tremendous value of their own.
- ⇒ We discussed results of previous studies and comparison with our work in the “Discussion” of the main text.

It is noteworthy that TMP has often been reported as an additive or co-solvent of the electrolyte for Mg-ion batteries.^{56, 57} These studies primarily focused on either modifying the solvation structure by incorporating TMP molecules in the solvation sheath, or influencing interfacial chemistry by constructing ion-conductive inorganic SEI layers rich in fluorides and phosphides. However, aspect such as moisture-tolerant manufacturing process for RMBs were not seriously addressed. In contrast, the present work focuses on developing a viable strategy to secure moisture tolerance through a simple and convenient surface activation process in TMP, thereby enabling dry room manufacturing of RMBs.

I have also following more specific comments.

1) I would disagree that moisture sensitivity of Mg metal anode is the key constraint preventing Mg battery market entry. I believe that many issues like developing suitable cathode materials improving long-term cycling performance and demonstrating high-energy density on practical cell level need to be addressed before. Hence, I would suggest rephrasing the abstract and introduction.

- ⇒ We are grateful to the reviewer for this valuable opinion. We totally agree with that many issues – such as developing suitable cathode materials, improving long-term cycling performance, and demonstrating high-energy density on practical cell level – remain urgent priorities to be addressed. It is also true that extreme moisture sensitivity of Mg metal poses a major obstacle to the development and commercialization of relevant technologies.
- ⇒ Therefore, we rephrased the abstract and introduction as follows.

Abstract ⁴

Despite striking advantages in terms of cost and safety, penetration of rechargeable Mg batteries (RMBs) into the commercial market is still hampered by major technical challenges including intrinsic hypersensitivity of Mg metal to moisture that readily forms a compact ion-insulating film on the surface. To unlock this critical constraint, a moisture-tolerant Mg electrode is developed that is capable of efficient Mg plating-stripping even in the highly moist electrolytes. Short immersion of Mg metal in trimethyl phosphate creates a sacrificial protection layer containing dimethyl magnesium in magnesium dimethyl phosphate that synergistically scavenges water molecules from the electrolytes instantly, enabling manufacturing of Mg-ion cells under moist and/or atmospheric conditions. This simple and scalable strategy provides a decisive breakthrough to reduce the manufacturing costs of RMBs, expediting their early commercialization.

In terms of commercial viability, however, this immense opportunity for the emerging battery market was hampered by several technical challenges in electrode materials and electrolytes including the extreme chemical vulnerability of Mg metals to moisture, where even a single exposure to a trace amount of water contaminant during the battery manufacturing process could be fatal to battery performance.^{32, 33} This is ascribed to the spontaneous formation of a

2) The authors confirm formation of $\text{Mg}(\text{DMP})_2$ on top of the Mg metal anode, which acts as scavenger during electrochemical cycling. It would be important to measure the water content in the wet electrolyte after cycling to evaluate scavenging effect during cycling. Did they also test several different electrolyte amounts in their cells.

⇒ We are grateful to the reviewer for this insightful comments and important supporting experiment suggestions. The reviewer suggested that measuring water content of the wet electrolyte after the electrochemical cycling would be a good indicator of confirming the scavenging effect. It would be ideal to measure the water content of the cycled electrolyte directly. However, since measuring water content using Karl-Fisher titration tools needs a sizable volume (at least 0.5 mL) of electrolyte, we tried alternative method as below instead of constructing large-volume-sized electrochemical cells.

- ⇒ We prepared two 0.5 mL of highly moist electrolytes containing about 3700 ppm and 6600 ppm of water content, respectively. 0.5 mL is the minimum volume with which moisture content was evaluated reliably with Karl-Fisher titration tools. In each of these electrolytes, five pieces of TMP-treated Mg metal discs were inserted and waited for three hours. This electrolyte volume corresponded to 100 μ L of electrolyte per TMP-treated Mg, which was the same with the volume applied to coin cell for electrochemical cycling.
- ⇒ As shown in Supplementary Table 2, the scavenged H₂O amount to 2100 ~ 2800 ppm. This indicates that TMP-treated Mg hold the capability of scavenging at least 2100 ~ 3000 ppm of moistures from the electrolyte within 3 hours.
- ⇒ This is a strong evidence that TMP-treated Mg scavenges moisture in the electrolyte effectively, lowering down moisture-level of the electrolyte system enough to enable reversible Mg plating-stripping.
- ⇒ We hope this alternative experiment clears the reviewer's concern.
- ⇒ In this study, Me₂Mg act as a main scavenging agent (chemical scavenging), while Mg(DMP)₂ plays a role as a subsidiary agent (physical scavenging).

Supplementary Table 2. Residual H₂O content in 0.5 mL of electrolyte measured by Karl-Fisher titration tools after immersing 5 pieces of TMP-treated Mg metal discs for 3 h.

Initial H ₂ O content / ppm	Final H ₂ O content / ppm	Amount of H ₂ O scavenged / ppm
3734.9	1622.4	2112.5
6606.4	3774.4	2832.0

- ⇒ In this study, the electrolyte amount applied was fixed at 100 μ L per each coin cell. We believe that this corresponded to the electrolyte-flooding condition. In lean electrolyte condition, TMP-treated Mg may work even under higher moist condition than 6600 ppm H₂O, since the absolute amount of water in the electrolyte would be smaller. Instead of trying different amount of electrolyte, we tried the electrolyte with the different amount of water content was investigated in this study. We trust that the reviewer will understand these circumstances.

3) The authors should showcase the effect of TMP also on better performing weakly coordinating electrolytes that have shown much better Mg plating/stripping electrochemical performance.^{3–5} Among them, MgAl(hfip)₄ has already showcased very high water tolerance.

- ⇒ We are grateful to the reviewer for proposing this interesting experiment involving the applicability of our strategy to the electrolytes with weakly coordinated anions. As the reviewer pointed out, the electrolytes based on the weakly-coordinated anions such as ethereal solution of Mg alkoxyborates and alkoxyaluminates, have been intensive research area in recent years. Among them, Mg[Al(hfip)₄]₂ in G2 electrolyte has already been shown to exhibit an excellent moisture-tolerance, enabling efficient Mg plating-stripping with up to 1000 ppm of water without a significant increase in overpotential (ACS Appl. Mater. Interfaces 2022, 14, 23, 26766–26774). To cope with the reviewer's request, we carried out cooperative studies with Prof. Zhao-Karger from KIT, Germany, who is one of the earliest developers of these electrolyte systems and provided us with Mg[B(hfip)₄]₂. Using salt they provided with, we prepared a series of electrolytes, **E4**, 0.3 M Mg[B(hfip)₄]₂ in G1 having moisture content of 2.0, 1092 and 3482 ppm H₂O, which are representative electrolytes among electrolytes with weakly coordinated anions. Unlike Mg[Al(hfip)₄]₂ in G2, the moisture-tolerance of ethereal solution of Mg[B(hfip)₄]₂ was not well studied so far.
- ⇒ We carried out Mg plating-stripping with a symmetrical configuration using TMP-treated Mg electrodes in **E4** electrolytes. The result is now shown in Supplementary Fig. 32 which indicates that TMP-treated Mg also worked in **E4** electrolytes under highly moist electrolyte having up to 3500 ppm H₂O, although moderate increase in the overpotential in Mg plating-stripping observed in the moist conditions. This proves that our strategy can work with Mg alkoxyborate electrolytes. These results were also confirmed by Zhao-Karger's group independently.
- ⇒ To emphasize the versatility of our strategy to various electrolyte systems, we included these experimental results and associated discussions in the "Discussion" of the main text.

⇒ In recognition of prof. Zhao-Karger's important contribution to revision of this paper, we added prof. Zhao-Karger, and Dr. Sibylle Riedel (who carried out the actual experiments) as co-authors of this paper.

Supplementary Fig. 32. Mg plating-stripping performance of symmetric cells with TMP-treated Mg electrodes in Mg[B(hfip)₄]₂-based E4 electrolyte containing different water contents. (A, C, E) Long-term cycling behavior using E4 electrolytes containing 2 ppm, 1092 ppm, and 3482 ppm of water, respectively. (B, D, F) Enlarged views of the first 20 cycles from (A), (C), and (E), respectively. The plating-stripping was carried out in 0.1 mA cm⁻², 0.1 mAh cm⁻².

Electrolytes based on weakly-coordinated anions, such as ethereal solution of Mg alkoxyborates and alkoxyaluminates, have attracted considerable research interest in recent years.^{31, 38, 39} In particular, Mg[Al(hfip)₄]₂ in G2 (hfip: 1,1,1,3,3,3-hexafluoroisopropoxy) has demonstrated efficient Mg plating-stripping performance even in the presence of 1000 ppm of water, without a significant increase in overpotential.³⁹ To access the compatibility of our strategy with such electrolytes, the reversibility of Mg plating-stripping was investigated in **E4** electrolyte: 0.3 M Mg[B(hfip)₄]₂ in G1, containing moisture levels of 2.0, 1092 and 3482 ppm H₂O, respectively – a system whose moisture tolerance has been relatively unexplored. With TMP-treated Mg electrodes, reversible Mg plating-stripping was still achievable even under highly humid conditions (up to ~3500 ppm H₂O), although a moderate increase in overpotential was observed (Supplementary Fig. 32). These results demonstrate the versatility of our strategy across various electrolyte systems.

4) More focus should be on the cycling in chloride-free electrolytes like **E3** since chloride-free electrolytes are the only realistic option for practical Mg batteries. Most of their cycling result are shown on the **E1** electrolyte, which contains MgCl₂ and it was already shown that MgCl₂-Mg(TFSI)₂ has better water tolerance than plain Mg(TFSI)₂. 6

- ⇒ We are grateful to the reviewer for the valuable comments. We agree that chloride-free **E3** electrolyte could serve as more favorable option for constructing practical Mg batteries. As the reviewer requested, we carried out more electrochemical and morphological characterization on various aspects of **E3** electrolyte.
- ⇒ At the same time, the main focus of this study is to introduce a new strategy for ensuring moisture-tolerance of Mg electrode for the purpose of realizing the construction of the practical RMBs that can be manufactured in a dry room facility.
- ⇒ Therefore, we decided to keep the original characterization format for **E1** and **E3** electrolytes, but added additional characterization data associated with **E3** electrolytes in Supplementary Figs. 12, 13, and 17. We also added full-cell performance of Mo₆S₈ with **E3** electrolytes (Supplementary Fig. 28).
- ⇒ Firstly, **E3** electrolyte was applied to symmetrical cell consisting of TMP-treated Mg electrodes to evaluate rate capability under varying moisture level as shown in Supplementary Fig. 12, which showed that TMP-treated Mg electrodes in moist **E3** electrolyte (200 ppm) worked as good as in the dry electrolyte (4 ppm).

Supplementary Fig. 12. Mg plating-stripping behavior under various current rates (0.1, 0.2, 0.5, 1, 2 mA cm⁻²) in the **E3** electrolytes with 4 and 200 ppm H₂O. The deposition was carried out for 1 h for each current rate.

⇒ We also observed morphological development of Mg metal deposits in **E3** electrolyte under the different moisture level. Mg metal deposits were obtained from asymmetrical configuration, Cu||TMP-treated Mg. The results are included in Supplementary Fig. 17. In dry electrolyte, Mg metal deposit was relatively uniform and round with particle size around 10 μm, while in moist electrolyte, Mg deposit turned into film-like morphology with much smaller particle size. The fact that Mg deposit could be obtained from the asymmetrical configuration proves that moisture scavenging mechanism by Me₂Mg and Mg(DMP)₂ from TMP-treated Mg electrode is effective in the **E3** electrolyte.

Supplementary Fig. 17. Morphological and elemental analyses on Mg deposits plated on Cu foil from the **E3** electrolytes with various moisture level, *e.g.*, (A) 4 ppm H₂O, and (B) 200 ppm H₂O. The plating was carried out in Mg||Cu asymmetric cells with 0.1 mA cm⁻², and 0.1 mAh cm⁻².

- ⇒ We also examined the reversibility of Mg plating-stripping with TMP-treated Mg electrode in **E3** electrolyte under high moisture level (200 ppm) depending on Mg deposition amount (areal capacity) (0.75~2 mAh cm⁻²). The results are shown in the Supplementary Fig. 13. We found that TMP-treated Mg electrode remained effective even under larger areal capacity deposition (2.0 mAh cm⁻²) at moist electrolyte condition, although the reversibility seemed to more distinct with **E1** electrolyte.

Supplementary Fig. 13. Mg plating-stripping behavior under various Mg areal capacities (0.75, 1, 2 mAh cm⁻²) in E3 electrolyte at (A) dry (4 ppm H₂O) and (B) moist (200 ppm H₂O) conditions. The current density applied was fixed at 0.1 mA cm⁻².

- ⇒ We also carried out additional full cell evaluation with Mo₆S₈ cathode, and E3 electrolyte containing different moisture level (4 vs. 200 ppm H₂O). The results are shown in Supplementary Fig. 28. When non-scraped Mg metal was applied to construct the full cell with E3 electrolyte, the cells did not work at all regardless of moisture content. But with TMP-treated Mg metal as an anode, the cell could be cycled normally in both dry and wet E3 electrolyte. They showed a similar discharge capacity and but increased overpotential during cycling compared to E1 electrolyte.

Supplementary Fig. 28. The discharge-charge profiles of the cells composed of Mo₆S₈ Chevrel phase cathodes and TMP-treated or scraped Mg metal anodes in dry (4 ppm H₂O) or moist (200 ppm) *E3* electrolyte: (A) TMP-treated Mg anode in dry *E3*; (B) scraped Mg anode in dry *E3*; (C) TMP-treated Mg anode in moist *E3*; (D) scraped Mg anode in moist *E3*. The cells were assembled in an argon-filled glovebox. The current rate was 0.05C (1C = 128 mA g⁻¹).

5) There is a lack of Mg metal interphase characterization during and after electrochemical cycling. I believe that surface characterization through electron microscopy, XPS, IR.... It could add important insight into the changes in the interphase during cycling.

- ⇒ We are grateful to the reviewer for the insightful and constructive comments. We agree with the reviewer on that it could be important to carry out post-mortem analysis of the protective layer that underwent the electrochemical cycling to elucidate the mechanism more clearly.
- ⇒ While this study was focused on scavenging H₂O molecules in the electrolyte or from the atmosphere during the manufacturing process in order to ensure the reversibility in

the subsequent cycles, post-cycling surface analysis of the electrode will clearly reveal the role of the protective layer.

- ⇒ For this purpose, we carried out 10 cycles of Mg plating-stripping in the **E1** electrolyte with TMP-treated Mg electrodes in symmetrical coin-cell configuration. After the cycling, the cells were dismantled inside Ar-filled glovebox and the electrodes were washed with plenty of dried G1. Then, the specimen was sent for SEM, XPS, and FT-IR analyses in a sealed container.
- ⇒ From the SEM images for the cycled surface, we found that the protective $\text{Mg}(\text{DMP})_2$ layer was still observed clearly and Mg deposit was mostly observed on the activated surface region of Mg electrode regardless of initial moisture level in the electrolytes (Supplementary Fig. 23). This indicates that moisture in the electrolyte was well eliminated by chemical scavenging agents in the protective layer, Me_2Mg . This also meant that Mg plating-stripping occurred typically in the protective surface region of Mg metal.

Supplementary Fig. 23. SEM images of TMP-treated Mg electrodes after 10 plating-stripping cycles in symmetric cells using **E1** electrolytes with 20 ppm, 374 ppm, and 1050 ppm of water. The dark regions indicate areas with limited reaction during TMP treatment, whereas the gray regions correspond to TMP-treated areas where smooth Mg deposition occurred, regardless of the initial moisture content in the electrolyte.

- ⇒ The comparison of XPS spectra of the TMP-treated Mg electrode before and after the electrochemical cycling indicated the overall XPS peaks were mostly not altered (Supplementary Fig. 24). Particularly, P 2p peak from $\text{Mg}(\text{DMP})_2$ was clearly observed in the cycled electrode. This indicates that the protective layer containing $\text{Mg}(\text{DMP})_2$ was well preserved after the electrochemical Mg plating-stripping cycling.

Supplementary Fig. 24. XPS spectra (survey scan) of TMP-treated Mg electrodes after 10 cycles of Mg plating-stripping in symmetric cells with *EI* electrolytes containing 1050 ppm of water. The presence of the P 2p peak in cycled TMP-treated electrode indicated that the $\text{Mg}(\text{DMP})_2$ protective layer was maintained after cycling.

- ⇒ FT-IR spectra were also obtained for TMP-treated Mg metal electrodes before and after the Mg plating-stripping cycling. The results are now shown in Supplementary Fig. 25. The comparison of the spectra indicated that $\text{Mg}(\text{DMP})_2$ component of the protective layer was still preserved even after the electrochemical Mg plating-stripping cycling.
- ⇒ All the post-mortem analyses on the cycled TMP-treated Mg electrodes indicated that the protective layer was relatively well preserved after the electrochemical Mg plating-stripping cycling. This implies that protective layers may facilitate uniform Mg deposition at later cycles in addition to their primary role as moisture-scavengers during cell assembly or initial electrochemical cycling stage. We appreciate to the reviewer for proposing this meaningful analysis to find out important secondary roles of protective layer.

Supplementary Fig. 25. FT-IR spectra of TMP-treated Mg electrodes before and after 10 cycles of Mg plating-stripping in symmetric cells. The spectra for $\text{Mg}(\text{DMP})_2$ was clearly observed in the cycled TMP-treated electrode, indicating that $\text{Mg}(\text{DMP})_2$ -based protective layer was retained after cycling.

Notably, the protective layers on Mg metal remained relatively intact throughout electrochemical cycling, implying that they may facilitate uniform Mg deposition in addition to their role as moisture-scavengers during cell assembly (Supplementary Fig. 23, Supplementary Fig. 24, Supplementary Fig. 25).

References:

- (1) Zhao, W.; Pan, Z.; Zhang, Y.; Liu, Y.; Dou, H.; Shi, Y.; Zuo, Z.; Zhang, B.; Chen, J.; Zhao, X.; Yang, X. Tailoring Coordination in Conventional Ether - Based Electrolytes for Reversible Magnesium - Metal Anodes. *Angewandte Chemie* 2022, 61 (30), e202205187. <https://doi.org/10.1002/ange.202205187>.
- (2) Li, C.; Guha, R. D.; Shyamsunder, A.; Persson, K. A.; Nazar, L. F. A Weakly Ion Pairing Electrolyte Designed for High Voltage Magnesium Batteries. *Energy Environ Sci* 2024, 17 (1), 190–201. <https://doi.org/10.1039/D3EE02861E>.
- (3) Mandai, T.; Youn, Y.; Tateyama, Y. Remarkable Electrochemical and Ion-Transport Characteristics of Magnesium-Fluorinated Alkoxyaluminate-Diglyme Electrolytes for

Magnesium Batteries. *Mater Adv* 2021, 2 (19), 6283–6296. <https://doi.org/10.1039/d1ma00448d>.

(4) Zhao-Karger, Z.; Gil Bardaji, M. E.; Fuhr, O.; Fichtner, M. A New Class of Non Corrosive, Highly Efficient Electrolytes for Rechargeable Magnesium Batteries. *J. Mater. Chem. A* 2017, 5, 10815–10820. <https://doi.org/10.1039/C7TA02237A>.

(5) Pavčnik, T.; Lozinšek, M.; Pirnat, K.; Vizintin, A.; Mandai, T.; Aurbach, D.; Dominko, R.; Bitenc, J. On the Practical Applications of the Magnesium Fluorinated Alkoxyaluminate Electrolyte in Mg Battery Cells. *ACS Appl Mater Interfaces* 2022, 14 (23), 26766–26774. <https://doi.org/10.1021/acsami.2c05141>.

(6) Connell, J. G.; Genorio, B.; Lopes, P. P.; Strmcnik, D.; Stamenkovic, V. R.; Markovic, N. M. Tuning the Reversibility of Mg Anodes via Controlled Surface Passivation by H₂O/Cl – in Organic Electrolytes. *Chemistry of Materials* 2016, 28 (22), 8268–8277. <https://doi.org/10.1021/acs.chemmater.6b03227>

Response to Reviewer #4's comments:

In this work the authors describe a chemical surface passivation approach using trimethyl phosphate (TMP) for stabilizing Mg metal electrodes to environmental processing and electrolytes with significant water content. Although this work is potentially of interest, TMP has been previously reported as an additive with beneficial properties for Mg batteries (Angew. Chem. Int. Ed. 2023, 62, e202304411). Therefore, more experimental data are needed about the specific mechanism by which this surface passivation strategy succeeds in order to more clearly define how their work differs from previous literature work in this system. Specific comments provided below:

- ⇒ We appreciate to the reviewer for the valuable opinion and constructive remarks. We were aware that TMP was previously reported in the literature as an electrolyte additive or co-solvent, for example, Angew. Chem. Int. Ed. 2022, 61, e202205187, Angew. Chem. Int. Ed. 2023, 62, e202304411 and ACS Energy Lett. 2025, 10, 1, 552–561. While TMP was utilized as an electrolyte additive in those studies, their main roles in their studies were either to modify the solvation structure of Mg^{2+} by introducing TMP molecule in the solvation sheath, or to influence interfacial chemistry by constructing ion-transporting inorganic SEI rich in fluorides and phosphides.
- ⇒ By contrast, in our work, TMP was NOT used as any kind of additive or solvent of the electrolytes. TMP was simply utilized as a surface activating agent for Mg metal anode. No TMP was added to the electrolytes when the electrochemical performance was evaluated in this study. Therefore, there is no intentional alteration in the solvation shell of Mg^{2+} , when the electrolytes like E1, E2 was applied to TMP-treated Mg electrodes. The purpose of TMP treatment of Mg metal in this study was to construct a moisture-tolerating protective film on the surface of Mg metal that could be applied to the cell assembly to enable dry room manufacturing process of RMBs. These purposes were not seriously dealt with in those studies. The resulting composite surface film consisted of moisture-scavenging Me_2Mg agents dispersed in $Mg(DMP)_2$ matrix. No TMP molecule existed in the composite protective film on Mg metal when it was applied to cell assembly. As such, while this study is distinct from the previous studies, it can be broadly categorized as being related to controlling Mg metal interphase.

- ⇒ Furthermore, prior studies have not examined the moisture tolerance in-depth and had not intended to develop inexpensive, scalable manufacturing process of RMBs. On the contrary, this study was much focused on developing a viable strategy for securing moisture tolerance through simple and convenient surface activation using TMP for the environmental manufacturing process of RMBs.
- ⇒ We discussed results of previous TMP studies and the comparison with our study in the “Discussion” of the main text.

It is noteworthy that TMP has often been reported as an additive or co-solvent of the electrolyte for Mg-ion batteries.^{56, 57} These studies primarily focused on either modifying the solvation structure by incorporating TMP molecules in the solvation sheath, or influencing interfacial chemistry by constructing ion-conductive inorganic SEI layers rich in fluorides and phosphides. However, aspect such as moisture-tolerant manufacturing process for RMBs were not seriously addressed. In contrast, the present work focuses on developing a viable strategy to secure moisture tolerance through a simple and convenient surface activation process in TMP, thereby enabling dry room manufacturing of RMBs. ↵

1. How much electrolyte was used to fill the cells used for testing? The surface film imaged in Figure S1 seems far too thin to effectively scavenge 1000s ppm of H₂O from the electrolyte, and quantitative analysis is necessary to explain how much Me₂Mg needs to form to accomplish this task. Analysis of the electrolyte water content after exposure to treated Mg surfaces is also needed to validate this takes place. For example, 6500 ppm H₂O corresponds to ~0.4 M, or approximately half of the total Mg²⁺ present in halide-containing electrolytes and almost the same as in halide-free electrolytes. This seems like far too much water to be accounted for just by scavenging by a surface film.

- ⇒ We are grateful to the reviewer for raising this issue. The amount of electrolyte applied to each coin cell was set to be 100 μL. We added the details in the Methods section.
- ⇒ We are aware of that the amount of electrolyte is a critical factor in quantifying the total water content in the cell. Accordingly, we evaluated whether M₂Mg from the TMP-

treated Mg can scavenge all the water in the electrolyte quantitatively. The results are now displayed in Table R-1 below.

Table R-1. Amount of Me_2Mg required to scavenge different levels of water in 100 μL of electrolyte. Values were calculated based on the 1:2 molar ratio of Me_2Mg reacting with H_2O .

H_2O in ppm	H_2O in μmol	Required Me_2Mg in μmol
30	0.201	0.101
300	2.012	1.006
1,000	6.706	3.353
3,000	20.12	10.06
6,500	43.59	21.79

- ⇒ From the analysis of cross-sectional image of TMP-treated Mg electrode, the average thickness of Mg metal reacted is around 10 μm . The corresponding amount of Mg metal is around 150 μmol per Mg disc of 16 mm diameter. According to the work of P. Becher (J. Am. Chem. Soc. 86 (1964) 1782), 10% of Mg metal reacted was transformed to Me_2Mg . Therefore, the amount of Me_2Mg in the protective film is estimated to be around 15 μmol per Mg disc. As shown in Table R-1, The amount of Me_2Mg necessary for scavenging water content in 100 μL of electrolyte containing 3000 ppm H_2O is around 10 μmol . This is less than the amount of Me_2Mg expected in TMP-treated Mg metal disc. Therefore, TMP-treated Mg electrode has the capability of scavenging at least 100 μL of electrolyte containing 3000 ppm H_2O (maximum capacity is estimated to be ~ 4500 ppm).
- ⇒ Meanwhile, the amount of Me_2Mg necessary for eliminating moisture from the electrolyte having 6500 ppm H_2O was estimated to be 22 μmol , which is greater than the amount of Me_2Mg (15 μmol) expected from the TMP-treated Mg disc. In this case near the edge of moisture-tolerance capacity, we believe that along with the chemical scavenging effect of Me_2Mg , $\text{Mg}(\text{DMP})_2$ might act as a good physical barrier to protect the Mg surface by adsorbing H_2O on its surface. Moreover, some fraction of H_2O might be adsorbed to glass wool separator in the cell as well.

- ⇒ It should be also noted that the expected moisture-tolerance capability in Table R-1 is an estimated value considering only Me_2Mg . Therefore, combination of several moisture-scavenging modes might constitute the final moisture-tolerance of TMP-treated Mg electrode.
- ⇒ Moreover, although P. Becher (1964) predicted that about 10% of Mg metal reacted was transformed to Me_2Mg , he also recognized that there might be significant amount of reactive species to humidity other than Me_2Mg from the reaction of Mg and TMP. Therefore, the percentage of reactive species could be more than 10% of Mg reacted from the reaction of Mg and TMP.
- ⇒ Measuring water content of the moist electrolyte after the electrochemical cycling could be a good indicator of confirming the scavenging effect of protective film. For this purpose, it would be ideal to measure the water content of the cycled electrolyte directly by Karl-Fisher titration tools. However, measuring water content reliably using Karl-Fisher titration tools needs a sizable volume (at least 0.5 mL) of electrolyte. Therefore, we tried alternative method as below instead of constructing large electrochemical cells.
- ⇒ We prepared two 0.5 mL of highly moist electrolytes containing 3700 ppm and 6600 ppm of water content, respectively. 0.5 mL is the minimum volume with which moisture content could be evaluated with Karl-Fisher titration tools. In each of these electrolytes, five pieces of TMP-treated Mg meta were inserted and waited for three hours, so that we simulated the actual electrochemical testing condition. This corresponded to 100 μL of electrolyte per TMP-treated Mg, which was the same with the volume applied to coin cell for electrochemical cycling.
- ⇒ As shown in Supplementary Table 2, the scavenged H_2O amounted to 2100 ~ 2800 ppm. This indicates that TMP-treated Mg hold the capability of scavenging at least 2100 ~ 3000 ppm of moistures from the electrolyte within 3 hours.
- ⇒ This is a strong evidence that TMP-treated Mg scavenges moisture in the electrolyte effectively, lowering down moisture-level of the electrolyte system enough to enable reversible Mg plating-stripping.
- ⇒ We hope this alternative experiment clears the reviewer's concern.
- ⇒ In this study, Me_2Mg act as a main scavenging agent (chemical scavenging), while $\text{Mg}(\text{DMP})_2$ plays a role as a subsidiary agent (physical scavenging).

Supplementary Table 2. Residual H₂O content in 0.5 mL of electrolyte measured by Karl-Fisher titration tools after immersing 5 pieces of TMP-treated Mg metal discs for 3 h.

Initial H ₂ O content / ppm	Final H ₂ O content / ppm	Amount of H ₂ O scavenged / ppm
3734.9	1622.4	2112.5
6606.4	3774.4	2832.0

2. Related to the point above – how does enough Me₂Mg survive exposure to air to still provide any benefit? This Grignard species should be fully reacted almost immediately when exposed to ambient conditions. Characterization after air exposure is needed to understand the surface chemistry of the treated Mg electrodes and confirm the same protection mechanism is active.

- ⇒ We are grateful to the reviewer for the insightful comments and raising this important and interesting issue. As the reviewer pointed out, we agree on that Me₂Mg may be decomposed instantly upon contact with moisture or oxygen gas from the air when it is exposed to air in normal circumstances. But in this study, we assembled the cells in a dry room facility, where dew point was maintained below -60 °C. Moreover, Grignard species, Me₂Mg were dispersed in the Mg(DMP)₂ matrix that might shield the infiltration of moisture and oxygen to some extent. Therefore, the decomposition rate would be much slower than the normal case.
- ⇒ To clarify this point, we carried out the additional experiments for evaluating the reversibility of Mg plating-stripping after TMP-treated Mg was exposed to dry room air in different period of time (5 min, 30 min, 1 h, 3 h). After set period of time, the cell was assembled within 1 min inside a dry room facility. This will give an informative clue whether our strategy can be applied to the actual environmental manufacturing process in a dry room facility.
- ⇒ The electrolyte containing about 1000 ppm of H₂O was applied to evaluate the moisture-tolerance after a span of exposure time to air. As shown in Fig. 5a, the

reversibility was relatively well-maintained up to 1 hour of exposure time. They lost moisture-tolerance apparently after 3 hours' exposure in the dry room atmosphere as we observed a sharp increase in the overpotential for Mg plating-stripping. This shows that scavenging agent (Me_2Mg) consistently eliminates moisture and oxygen content in a cell during or after the exposure to air environment.

- ⇒ We believe that one hour's moisture tolerance is enough to assemble the battery cells in a dry room facility. This indicates that our strategy can be applied to the actual environmental manufacturing process in a dry room facility.

Fig. 5. Characterization of TMP-treated Mg electrodes pre-exposed to dry room air for different time period.

(a) Mg plating-stripping behavior of symmetrical cells in *EI* (997 ppm H_2O) composed of a pair of TMP-treated Mg electrodes, which was pre-exposed to dry room air for 5 min., 30 min., 1 h, and 3 h. (b) GC-FID spectra for the gases that evolved after the TMP-treated Mg electrode, which were pre-exposed to dry room air for 0 min., 5 min., 30 min., and 3 h, respectively, were dipped in moist electrolyte (*EI*, 997 ppm H_2O).

- ⇒ In order to find more direct evidence that Me_2Mg survived the air exposure in the dry-room environment for some time, we carried out the additional experiments for speciating the gas through GC-FID from the reaction between moist electrolyte (997 ppm H_2O) and TMP-treated Mg metal, which was pre-exposed to dry room air in different period of time (5 min, 30 min, 3 h). The results are shown in Fig. 5b, which showed that even after 30 min, the distinct peak for methane gas was clearly observed, although the intensity for methane decreased gradually over exposure time. After 3 hours of exposure, no peak for methane gas was observed, indicating most of Me_2Mg

was decomposed by that time. This also means that TMP-treated Mg lost the chemical scavenging capability by that time. This agrees with the electrochemical observation (Mg plating-stripping) that they developed an excessive overpotential for TMP-treated Mg electrodes which was exposed to air for 3 hours in the dry room facility prior to the test. Therefore, Me_2Mg can survive at a meaningful population level for up to about one hour after being exposed to the air in the dry-room facility.

⇒ We believe that one hour's moisture tolerance is enough to assemble the battery cells in a dry-room facility. This indicates that our strategy can be applied to the actual environmental manufacturing process in a dry room facility.

3. The primary reaction mechanism proposed with Me_2Mg results in $\text{Mg}(\text{OH})_2$ formation. This is insoluble in ethers, and so it should still passivate the Mg surface. How then is Me_2Mg protecting the surface without simultaneously passivating it?

⇒ We are grateful to the reviewer for the insightful comments. We totally agree with the reviewer's opinion that once even thin layer of $\text{Mg}(\text{OH})_2$ were formed on Mg surface, it could passivate the Mg surface to block further ion transport.

⇒ Firstly, we would like to emphasize that Grignard reagent-like species, Me_2Mg is highly soluble in ethereal solution including G1, and G2 used in this work. Therefore, it is highly likely that as soon as electrolyte is placed in contact with TMP-treated Mg electrode, Me_2Mg will permeate quickly into the electrolyte to scavenge H_2O molecules instantly, generating $\text{Mg}(\text{OH})_2$ particles and methane gas in the bulk electrolyte, not on the Mg surface. We believe that these $\text{Mg}(\text{OH})_2$ particles will remain either suspended in the electrolyte or attached to hydrophilic moieties of $\text{Mg}(\text{DMP})_2$ layer, or glass wool separators.

⇒ We put TMP-treated Mg and scraped Mg in the moist G1 solvent (3135 ppm H_2O) for 1 day, respectively. Then the solvent was analyzed to check the content of Mg species in the electrolyte. We found 1.33 ppm for the solvent with scraped Mg and 65.1 ppm for the solvent with TMP-treated Mg. This indicate that $\text{Mg}(\text{OH})_2$ was created, being suspended in the solvent from the reaction between moisture and Me_2Mg .

- ⇒ It is also noteworthy that the amount of Me_2Mg in the protective layer is regarded to be high enough to convert at least 4500 ppm of water completely to $\text{Mg}(\text{OH})_2$.
- ⇒ The fact that reversible Mg plating-stripping occurred on TMP-treated Mg or SS working electrode is self-explanatory that no compact $\text{Mg}(\text{OH})_2$ layer existed on Mg metal surface.

Supplementary Table 1. Mg content suspended in the solvent measured by ICP-OES after 5 pieces of TMP-treated or scraped Mg metal discs were immersed in 5 mL of moist G1 with 3135 ppm H_2O for 1 day.

Mg metal in moist G1 (3135 ppm H_2O)	Mg content / ppm
Scraped Mg	1.33
TMP-treated Mg	65.1

4. How was the Mg dimethyl phosphate (DMP) prepared? It is not described in the methods and this is important to help validate the proposed formation of Me_2Mg . Extrinsic reaction of $\text{Mg}(\text{DMP})_2$ prior to water exposure could also explain the lack of methane evolution observed in Figure 3b. A more direct demonstration of Me_2Mg formation would be preferable, either by additional spectroscopic analysis or other methodologies, as its presence is central to the proposed mechanism of protection.

- ⇒ We are grateful to the reviewer for pointing out this matter. The synthesis of $\text{Mg}(\text{DMP})_2$ was carried out by following the known procedure for the synthesis for NaDMP with some modification. The information is now supplemented in the Methods section of paper as follows.

Synthesis of Mg(DMP)₂. To synthesize Mg(DMP)₂, 4 mmol of TMP (0.43 mL) was mixed with 2 mmol of MgI₂ (0.507 g) in G1 (20 mL) by following a previously reported synthesis of NaDMP.⁶¹ The solution was stirred at 50 °C for 3 days in an Ar-filled glove box. Then, the resulting product was filtered and dried under vacuum. The FT-IR spectrum of Mg(DMP)₂ was similar to that of NaDMP prepared by previously reported method, confirming the successful synthesis of Mg(DMP)₂ (Supplementary Fig. 33).

- ⇒ The whole synthetic process for Mg(DMP)₂ was carried out in an Ar-filled glovebox with strict environment control, so that Mg(DMP)₂ was not exposed to any moisture or oxygen prior to water exposure or other analyses. We can confirm the successful synthesis of Mg(DMP)₂ by comparison of FT-IR spectra with NaDMP (Supplementary Fig. 33).

Supplementary Fig. 33. FT-IR spectra of synthesized Mg(DMP)₂ and NaDMP.

- ⇒ In order to examine the reactivity of Mg(DMP)₂ with moist in the electrolyte, Mg(DMP)₂ was put into the water, and was subject to H-NMR analysis. No sign of possible decomposed products was observed from H-NMR spectra (Supplementary Fig. 19). From these observations, we concluded that Mg(DMP)₂ does not react with water and the methane evolution in Fig. 3b did not originate from the decomposition of Mg(DMP)₂

with water. This provides a strong basis for interpreting the CH₄ evolution observed in Fig. 3b as originating from Me₂Mg.

- ⇒ The formation of Me₂Mg from the reaction of TMP and Mg metal was also reported in the previous literature (J. Am. Chem. Soc. 74, 2923-2924 (1952)). But, as reviewer also mentioned, the strong reactivity of dimethyl magnesium with environmental humidity and air made it difficult to analyze its presence.
- ⇒ To obtain the more direct spectroscopic evidence of the existence of Me₂Mg in the protective layer, TMP-treated Mg metal was put in the ultra-dried THF (and G1) solvent so that Grignard species, Me₂Mg, diffuses into the THF (and G1) solution in a argon-filled glovebox. Then, the filtrate solution was subject to IR spectroscopy in the absolute inert atmosphere. The results are now shown in Supplementary Fig. 18. The absorption peak at ~500 cm⁻¹ could be assigned to asymmetric stretching mode of C-Mg-C bond of Me₂Mg as reported in the literature (R. Salinger and H. Mosher, J. Am. Chem. Soc. 1964, 86, 9, 1782-1786). Therefore, it is highly likely that Me₂Mg was present in the protective layer.
- ⇒ But, after immersion of TMP-treated Mg in the highly moist solvent, the absorption band at ~500 cm⁻¹ disappeared, indicating the Me₂Mg reacted with moisture to convert into methane gas and Mg(OH)₂.
- ⇒ These observations support the conclusion that methane evolution in Fig. 3b arises from the reaction of Me₂Mg with water, rather than from Mg(DMP)₂ with water.

Supplementary Fig. 18. FT-IR spectra of the (A) THF and (B) G1 solution before and after immersion of a piece of TMP-treated Mg disc for 10 min. Each solvent was prepared in an ultra-dry (< 10 ppm H₂O) and moist (~6500 ppm H₂O) conditions. The absorption peak at ~500 cm⁻¹ could be assigned to asymmetric stretching mode of C-Mg-C band of Me₂Mg as reported in the literature.¹ Therefore, after immersion in the solvent, Me₂Mg released from the TMP-treated Mg is expected to remain dissolved in the dried solvent, whereas it is unlikely to survive in a highly moist solvent.

5. How do the authors rule out hydrogen evolution/oxidation as the primary electrochemical couple in two electrode cells with hundreds to thousands of ppm H₂O? It is thermodynamically preferred to Mg deposition/stripping and water is present at high enough concentrations to significantly contribute to the measured current. Additional electrochemical experiments are needed, ideally with a three-electrode setup to validate the potentials at which the electrodes are operating, in order to confirm that the observed electrochemistry is actually related to Mg plating/stripping.

⇒ We are grateful to the reviewer for these insightful comments and suggesting supporting experiments. We agree with reviewer on that water molecules may participate in the electrochemical reactions in two- electrode cells and contribute to the measured current significantly when their concentrations are as high as hundreds to thousands of ppm H₂O. This would be true indeed, unless water in the moist electrolyte is scavenged, and its concentration is sufficiently lowered down. But, in this study, most of water present in the electrolyte initially was eliminated by moisture-scavengers,

Me₂Mg or Mg(DMP)₂, in the protective layer before cell starts. Therefore, the contribution from the water decomposition during cycling is not considered significant. We confirmed that moisture level in the electrolyte can be significantly lowered via moisture-scavenging by TMP-treated electrode from Supplementary Table 2.

⇒

Supplementary Table 2. Residual H₂O content in 0.5 mL of electrolyte measured by Karl-Fisher titration tools after immersing 5 pieces of TMP-treated Mg metal discs for 3 h.

Initial H ₂ O content / ppm	Final H ₂ O content / ppm	Amount of H ₂ O scavenged / ppm
3734.9	1622.4	2112.5
6606.4	3774.4	2832.0

- ⇒ If the concentration of water in the electrolyte exceeded the moisture-tolerance of TMP-treated Mg electrode at the beginning, it would lead to the active participation of water molecules in the electrode reactions. This will eventually result in the formation of Mg(OH)₂ passivation layer on Mg electrode as reviewer pointed out in other comments. In this case, the electrochemical cycling will be overwhelmed by excessive overpotential development just like Supplementary Fig. 5.
- ⇒ Three electrode setup to validate the potentials at which the electrodes are operating, is also doomed to have a challenging problem with water contamination, because water molecules can slowly permeate into the reference side through its porous plug, causing complicated mixed potential in the reference electrode, making it difficult to record the accurate electrode potential just like two-electrode setup.
- ⇒ The typical efficiency for the galvanostatic cycling with TMP-treated Mg metal/SS asymmetrical configuration in the moist electrolyte is more than 98% (Supplementary Fig. 14). This means that measured current can be mostly attributed to reversible Mg plating-stripping, since HER on Mg metal is not considered reversible. On the contrary, the efficiency for the galvanostatic cycling with untreated Mg metal/SS asymmetrical configuration in the moist electrolyte is close to 0% (Supplementary Fig. 15). This

indicates that water in the electrolyte participated actively in the electrode reaction to create passivation layer before any Mg plating-stripping occurs.

- ⇒ From the morphological observations of the plated electrode (in asymmetrical cell) in the moist electrolyte, Mg metal was clearly confirmed from SEM-EDS analysis, indicating that Mg plating was mainly responsible for the current signal measured. Moreover, the fact that the electrochemical profiles for Mg plating-stripping was similar regardless of initial moisture level in the electrolyte and that the full cell (with TMP-treated Mg metal) exhibited similar voltage profiles regardless of initial moisture level, implies that most of waters in the moist electrolyte was eliminated by moist-scavengers.
- ⇒ These observations indicate that moisture in the electrolyte is efficiently removed by scavenging mechanism installed in the protective layer of TMP-treated Mg metal. Therefore, in this study, the influence of moisture on the electrochemical reaction is considered highly limited when TMP-treated Mg electrode is used.

We would like to again express our sincere appreciation to all the reviewers for their valuable and insightful comments. We hope that we have adequately addressed all the concerns raised. We believe that our manuscript has significantly improved thanks to the reviewers' **constructive and thoughtful** comments.

Si Hyung Oh, PhD

Principal investigator and professor

Energy Storage Research Center, Korea Institute of Science Technology,

5, Hwarang-ro 14-gil, Seongbuk-gu, Seoul 02792, Republic of Korea,

Email- sho74@kist.re.kr

Table of Contents

1. Response to Reviewer #3's comments.....	2
2. Response to Reviewer #4's comments.....	4

Response to Reviewer #3's comments:

The authors have adequately addressed most of my concerns and manuscript should be suitable for publication after minor revision.

- ⇒ We are deeply grateful for the reviewer's thoughtful and constructive comments, which have been invaluable in refining and strengthening this work

Nevertheless, I would like to authors to comment on the increased overpotential of MgBhfip-based electrolytes (E4) with increasing water content, as well as in E3. Does this mean that there is still a beneficial effect of chloride species in wet electrolytes? Where they able to identify chloride-based species in analysis of their cycled electrodes?

- ⇒ We thank the reviewer for this valuable comment. The increased overpotential observed in E3 and E4 with higher water content can be attributed to the formation of thin passivation films on the Mg surface, caused by residual water that was not completely scavenged. Such films may hinder Mg^{2+} transport across the interface and increase the kinetic barrier for plating and stripping.
- ⇒ In the chloride-containing electrolytes like E1 and E2, the presence of chloride anions appears to partially mitigate this effect, as TMP-treated Mg electrode endure higher moisture content in chloride-containing E1 (~6500 ppm H_2O) compared to chloride-free E3 (~4500 ppm H_2O). As reported in previous studies, Cl^- is highly effective in breaking down the passivation films formed on some metal surface, triggering pitting corrosion (J. Muldoon et al., Energy Environ. Sci. 2013, 6, 482). Therefore, it is plausible that chloride-containing Mg-Cl-based complexes weaken or locally disrupt the compact passivation layer, thereby allowing more reversible Mg plating and stripping even in the presence of trace moisture.
- ⇒ The direct detection of chloride-containing species on the surface of cycled electrodes was limited (Cl^- species was not detected on cycled TMP-treated Mg metal from XPS analysis), probably due to high solubility of Cl-containing species in the electrolyte.

⇒ A brief comment of this point has been added to the revised manuscript as below.

overpotential was observed (Supplementary Fig. 33). This likely arises from thin passivation films formed by residual moisture, whereas chloride-containing electrolytes showed better moisture tolerance and low overpotential due to chloride-induced disruption of these films. These results demonstrate the versatility of our strategy across various

Response to Reviewer #4's comments:

The authors have satisfactorily addressed several of my initial concerns; however, two key issues must still be addressed before this manuscript can be considered further:

⇒ We sincerely appreciate the reviewer for the insight and constructive feedback, which has significantly contributed to improving our manuscript. A detailed, point-by-point response to each comment is provided below.

1. The measurements of H₂O and Mg content before and after electrolyte exposure to treated Mg foils are informative, but they do not support a key conclusion of the manuscript. Specifically, the authors hypothesize that Mg(OH)₂ formed from reaction of Me₂Mg with H₂O remains suspended in the electrolyte, but that cannot be the case. The removal of 2000-3000 ppm of H₂O by Me₂Mg should quantitatively result in the formation of 1000-1500 ppm of Mg(OH)₂. However, only 65 ppm Mg is observed in the electrolyte after reaction. This is two orders of magnitude less than would be expected. Where does the Mg(OH)₂ go if it's not on the Mg metal? As I mentioned in my original review, understanding of the protection mechanism is key to differentiating this approach from other work on this system, and these results leave significant ambiguity as to what is actually happening.

⇒ We sincerely thank the reviewer for the insightful comments and for raising this important point again. It is true that complete removal of 2000~3000 ppm of H₂O (average ~2500 ppm) from the electrolyte by Me₂Mg should theoretically produce 1000~1500 ppm of Mg(OH)₂, while only 65.1 ppm of Mg was detected in the electrolyte after the reaction (from the ICP analysis). We fully understand the reviewer's concern. However, the apparent discrepancy can be rationalized by the following considerations.

⇒ In brief, the primary reason for this discrepancy originated from that for ICP analysis, 5 mL of moist electrolyte was used with five TMP-treated Mg discs (thus 0.5 mL per one TMP-treated Mg disc), whereas in the Mg||Mg symmetrical cell test, 0.05 mL of electrolyte per TMP-treated Mg electrode was used. The total amount of water present in 0.5 mL of moist electrolyte (for ICP analysis) should be ten times higher than that in

0.05 mL (for electrochemical test). This difference, together with partial removal of H₂O by physical scavenging (*i.e.*, adsorption on the Mg(DMP)₂ layer), accounts for part of the discrepancy. We would like to explain these in detail as below.

- ⇒ To describe this quantitatively, we introduce the concept of ‘scavenging power’ of a TMP-treated Mg electrode, defined as the number of micromoles of H₂O removed by a single TMP-treated Mg disc. This scavenging power consists of two contributions: (i) **chemical scavenging**, arising from the reaction of Me₂Mg with H₂O (R-1 in the main text), and (ii) **physical scavenging**, arising from H₂O adsorption on the Mg(DMP)₂ layer.
- ⇒ In Karl-Fischer titration, the minimum electrolyte volume required was 0.5 mL. Five TMP-treated discs were immersed to simulate the asymmetric cell condition in coin cell (0.1 mL of electrolyte per Mg disc). Given that 2500 ppm H₂O in 0.1 mL corresponds to ~14 μmol of H₂O, this implies that one TMP-treated Mg disc exhibits a total scavenging power of approximately 14 μmol H₂O.
- ⇒ For ICP measurement, **5 mL** of electrolyte and five TMP-treated Mg discs were used. The detected Mg concentration was 65.1 ppm, corresponding to about **13.6 μmol**. Based on reaction (R-1), Me₂Mg + 2H₂O => Mg(OH)₂ + 2CH₄, this corresponds to **27.1 μmol** H₂O consumed, or **5.4 μmol** per TMP-treated Mg disc (that is, scavenging power). Considering that a portion of Mg(OH)₂ may remain adsorbed on the electrode surface and not detected in the ICP supernatant, the actual chemical scavenging power is likely higher. Thus, the chemical contribution accounts for roughly 39% of the total observed scavenging power.
- ⇒ To examine the role of physical scavenging (moisture trapping by Mg(DMP)₂), we prepared Mg(DMP)₂ powder and measured its ability to remove moisture from the electrolyte (Supplementary Table 3). Specifically, 0.413 g of Mg(DMP)₂ (the estimated amount that would exist in five TMP-treated Mg discs in previous revision) was added to 0.5 mL of moist G1 electrolyte (3491.3 ppm). After standing for 3 h, the H₂O content decreased by 662.9 ppm (by Karl-Fisher titration), correspond to a scavenging power of ~**3.7 μmol** per TMP-treated Mg disc. This represents about 27% of the total observed scavenging power.

- ⇒ By combining chemical (5.4 μmol) and physical (3.7 μmol) scavenging, the total scavenging capability of one TMP-treated Mg disc is estimated to be **$\sim 9.1 \mu\text{mol}$** , which explains approximately 66% of the total observed H_2O removal. This corresponds to removal of $\sim 1640 \text{ ppm}$ H_2O when 0.1 mL of electrolyte is used in a coin cell. Moreover, in case of a Mg||Mg symmetrical cell containing two TMP-treated Mg electrodes, the H_2O reduction capacity would increase to $\sim 3280 \text{ ppm}$, in a reasonable agreement with experimental observations.
- ⇒ We acknowledge that physical scavenging plays a role as important as chemical scavenging in the overall electrolyte dehydration mechanism. We added a clarification in the revised main text to emphasize this contribution.
- ⇒ We sincerely thank the reviewer again for prompting us to recognize and articulate this important aspect of our mechanism.

To assess the contribution of physical scavenging to the overall dehydration process, we examined the role of the $\text{Mg}(\text{DMP})_2$ layer formed on the TMP-treated Mg surface. This layer can adsorb residual moisture from the electrolyte, as supported by a separate test showing that synthesized $\text{Mg}(\text{DMP})_2$ powder reduced the H_2O content of a moist electrolyte by approximately 660 ppm (Supplementary Table 3). These results indicate that the $\text{Mg}(\text{DMP})_2$ layer provides an auxiliary pathway for moisture removal, complementing the chemical scavenging of Me_2Mg .

Supplementary Table 3. Measurement of residual H_2O content in 0.5 mL of moist electrolyte after addition of 0.413 g $\text{Mg}(\text{DMP})_2$ powder for 3 h by Karl-Fischer titration.

Initial H_2O content / ppm	Final H_2O / ppm	Amount of H_2O scavenged / ppm
3491.3	2828.4	662.9

2. Three electrode measurements of Mg plating/stripping in water-containing electrolytes are required to validate the proposed mechanism and confirm that Mg plating/stripping is

the primary electrochemical process being driven in these electrolytes after exposure to treated metal. As the authors note, there are other species introduced into the electrolyte by contact with the treated electrodes, and there is still significant residual water that remains after reaction (>1000 ppm). Any or all of these species can participate in electrochemical side reactions that will obscure inefficiency in a two electrode measurement, even in an asymmetric cell configuration. Two electrode galvanostatic measurements cannot convincingly rule out side reactions. The authors' claims of practical difficulty in performing three electrode measurements (e.g., water inclusion from the reference electrode) are inconsistent with an extensive body of literature on non-aqueous Mg electrochemistry. Three electrode measurements are quite possible, and they have been demonstrated in systems with Grignard reagents (e.g., 10.1038/35037553) and where water was added intentionally (e.g., 10.1021/acs.chemmater.6b03227). This point again speaks to the need to better define the protection mechanism by ruling out the possibility of side reactions that contribute to the electrochemistry being probed.

- ⇒ We thank the reviewer for raising this important issue again and for the helpful references regarding three-electrode measurements.
- ⇒ As requested, we successfully performed three-electrode electrochemical tests using 1/2-inch PTFE Tee-union fitting. A commercial Ag/AgCl reference electrode was modified to fit into PTFE Tee, and E1 electrolyte (MgTFSI₂/MgCl₂/DME) was used as the filling solution to minimize the liquid-junction potential with the bulk E1 electrolyte. A TMP-treated Mg disc or pristine Mg disc was used as the counter electrode, while a stainless steel disc (12 mm diameter, 1.1 cm² area) served as the working electrode. The electrolyte volume was 0.2 mL to ensure stable reference contact.
- ⇒ Because the three-electrode cell used twice the electrolyte volume (0.2 mL vs. 0.1 mL), half the electrode area (1.1 cm² vs 2.0 cm²), and only one TMP-treated Mg electrode as a counter electrode (instead of two in Mg||Mg cell in coin cell configuration), the effective moisture tolerance corresponds to roughly one-eighth of that in the coin-cell configuration. In other words, 100 ppm H₂O in the tree-electrode cell is approximately equivalent to 800 ppm in the Mg||Mg symmetric coin cell.

- ⇒ We first validated the modified Ag/AgCl/E1 reference electrode using ultra-dry electrolyte (< 10 ppm H₂O, equivalent to ~80 ppm in symmetric coin-cell scale) and pristine Mg counter electrode. Cyclic voltammetry exhibited clear Mg plating/stripping redox potential at -2.22 V vs. Ag/AgCl (Fig. 5c), which are well consistent with the literature values (~-2.2 V vs. Ag/AgCl, N. M. Markovic et al., Chem. Mater. 2016, 28, 8268) for the Mg/Mg²⁺ redox couple. Because the electrolyte contained negligible water, these peaks are unambiguously assigned to Mg plating and stripping.
- ⇒ Next, we examined moist electrolytes in the same three-electrode setup (with pristine Mg counter electrode). After 3 h's resting period to allow moisture scavenging, the current associated with Mg plating-stripping decreased drastically to less than 1% of that observed for the 15 ppm electrolyte (~120 ppm H₂O in coin-cell scale) (Supplementary Fig. 32B). For the electrolyte containing 40 ppm H₂O (~ 320 ppm H₂O in coin-cell scale), almost no current response was detected, indicating formation of a passivating layer on Mg metal consistent with previous observations.
- ⇒ When the TMP-treated Mg disc was used as the counter electrode, distinct Mg plating/stripping peaks (at -2.24 V vs. Ag/AgCl/E1) appeared not only in the ultra-dry electrolyte, but also in the moist electrolytes (Fig. 5d). The current magnitude was comparable to that in the dry electrolyte, and the redox potential (-2.22 V vs Ag/AgCl/E1) remained almost unchanged, confirming that the observed current corresponds to the Mg/Mg²⁺ process. No additional peaks were observed between -3.0 V and 0.0 V (vs. Ag/AgCl/E1), excluding hydrogen evolution or oxidation reactions. The plating-stripping efficiency in moist electrolyte was nearly identical to that in the dry electrolyte (83.2% vs. 80.9%, respectively). Importantly, reversible Mg plating was sustained up to ~ 491 ppm H₂O in the three-electrode cell, equivalent to ~ 4000 ppm H₂O in the Mg||Mg coin-cell configuration. These results clearly demonstrate that TMP-treated Mg effectively scavenges moisture from the electrolyte to a level sufficient for reversible Mg plating/stripping.
- ⇒ A brief description has been added to the Discussion section, based on the three-electrode experiments suggested by the reviewer, to clarify that the current response observed in the moist electrolyte with TMP-treated electrodes originates from Mg plating/stripping, not from hydrogen evolution or oxidation processes.

⇒ We sincerely thank the reviewer for suggesting the three-electrode experiments. This constructive suggestion allowed us to improve our study and substantially enhance the quality and clarity of the manuscript.

To further confirm that the electrochemical process observed in moist electrolytes corresponds to Mg plating and stripping, additional three-electrode measurements were carried out using a modified Ag/AgCl reference electrode⁴⁷,⁵⁶. The cyclic voltammograms (Fig. 5c, 5d, Supplementary Fig. 32) clearly showed reversible Mg plating/stripping peaks even in moist electrolytes when TMP-treated Mg was used as the counter electrode, while no additional peaks attributable to hydrogen evolution or oxidation reactions were detected. Notably, the Mg/Mg²⁺ redox potential remained nearly identical between dry and moist electrolytes (around -2.2 V vs. Ag/AgCl), indicating that the electrochemical process was not influenced by moisture-induced side reactions. These results confirm that the current response in the moist electrolyte originates from Mg plating/stripping rather than from side reactions, thereby supporting the proposed mechanism.⁴⁷

Fig. 5. Characterization of TMP-treated Mg electrodes pre-exposed to dry-room air for different time period and verification of Mg redox potentials via cyclic voltammetry. (a) Mg plating-stripping behavior of symmetrical cells in *E1* (997 ppm H₂O) composed of a pair of TMP-treated Mg electrodes, which was pre-exposed to dry room air for 5 min., 30 min., 1 h, and 3 h. (b) GC-FID spectra for the gases that evolved after the TMP-treated Mg electrode, which were pre-exposed to dry room air for 0 min., 5 min., 30 min., and 3 h, respectively, were dipped in moist electrolyte (*E1*, 997 ppm H₂O). **Cyclic voltammetric curves obtained from three-electrode cells with Ag/AgCl as the reference electrode, (c) a scraped Mg disc as the counter electrode (CE), and (d) a TMP-treated Mg disc as the CE in *E1* electrolyte containing various water contents. The moisture tolerance in the three-electrode setup is about one-eighth that of the Mg||Mg coin cell (see Materials and Methods for details).**

Supplementary Fig. 32. Cyclic voltammetry (CV) curves of the three-electrode systems. The measurements were conducted with a stainless steel foil working electrode (WE). The reference electrode (RE) was Ag/AgCl immersed in dried *EI* electrolyte. (A-C) CV profiles obtained using a scraped Mg counter electrode in *EI* electrolyte containing various water contents. (D-F) CV profiles obtained using a TMP treated Mg counter electrode in *EI* electrolyte containing various water contents. The inset in (A, D, E) show the corresponding Coulombic Efficiency (CE) of the systems.

Other questions:

1. The authors mention in their response that Me₂Mg would be expected to react with O₂, but that performing work in a dry room prevents this reactivity. The amount of O₂ in the atmosphere of a dry room is not significantly different than in a regular laboratory space. Please clarify this point – if Me₂Mg reacts with O₂, how does it survive the dry room environment? Does some other species form that still results in the release of CH₄ when exposed to H₂O?

⇒ We appreciate the reviewer's insightful comment. It is indeed correct that the O₂ concentration in a dry room is comparable to that in normal laboratory air. We would like to clarify that we did not intend to suggest that Me₂Mg does not react with O₂ in a dry room. We fully agree that Me₂Mg readily reacts with both O₂ and H₂O in a dry room

in the same way as it does in normal laboratory air. Our intention was not to claim that this reaction is prevented in a dry room, but rather to investigate how effectively Me_2Mg can scavenge the small residual amounts of O_2 and H_2O that remain under realistic dry-room conditions.

- ⇒ Our interest here lies in evaluating whether TMP-treated Mg electrodes can enable MIB manufacturing processes within a dry room facility. For this purpose, the electrodes must be able to block or scavenge trace H_2O and O_2 before a compact passivation layer forms on the Mg surface. Performing battery assembly in normal laboratory air is impractical for both MIBs and LIBs because of the high and continuously replenished moisture content. Therefore, our aim was to assess whether MIB cell assembly could be implemented in a dry room instead of a glovebox filled with expensive high-purity inert gas.
- ⇒ As shown in Fig. 5a, reversible Mg plating/stripping was achieved even after the TMP-treated Mg electrodes were exposed to dry-room air for up to 1 h. During this period, residual O_2 and H_2O in the dry-room atmosphere is expected to be continuously scavenged by Me_2Mg within the TMP-treated layer, while the protective film simultaneously serves as a physical barrier that impedes further infiltration of these species. A 1-hour window is sufficient to fabricate a typical battery cell. We therefore believe that TMP-treated Mg electrodes represent a key technology for realizing dry-room manufacturing of MIBs, which would be an important step toward their practical commercialization.

Fig. 5. Characterization of TMP-treated Mg electrodes pre-exposed to dry-room air for different time period and verification of Mg redox potentials via cyclic voltammetry. (a) Mg plating-stripping behavior of symmetrical cells in *E1* (997 ppm H₂O) composed of a pair of TMP-treated Mg electrodes, which was pre-exposed to dry room air for 5 min., 30 min., 1 h, and 3 h. (b) GC-FID spectra for the gases that evolved after the TMP-treated Mg electrode, which were pre-exposed to dry room air for 0 min., 5 min., 30 min., and 3 h, respectively, were dipped in moist electrolyte (*E1*, 997 ppm H₂O). Cyclic voltammetric curves obtained from three-electrode cells with Ag/AgCl as the reference electrode, (c) a scraped Mg disc as the counter electrode (CE), and (d) a TMP-treated Mg disc as the CE in *E1* electrolyte containing various water contents. The moisture tolerance in the three-electrode setup is about one-eighth that of the Mg||Mg coin cell (see Materials and Methods for details).

2. Beyond the appearance of the peak at 500 cm⁻¹ in dry solvents exposed to treated Mg electrodes, significant changes to both the THF and DME vibrational modes are also observed in the FTIR spectra in Figure S18. What is the origin of these changes (e.g., Me₂Mg coordination with the solvents, decomposition product formation, some other effect)? Additional discussion is needed to explain these changes. A more convincing experiment would be to add water to the dry solvents after exposure to treated Mg to see if the spectra return to what is observed for the unexposed solvents.

⇒ We appreciate the reviewer's valuable comment and the suggestion for additional experiments. Most of the peaks beyond 500 cm^{-1} in the FT-IR spectra of dry solvents exposed to the treated Mg electrodes correspond to the characteristic bands of $\text{Mg}(\text{DMP})_2$, as shown in Fig. R1. We found that $\text{Mg}(\text{DMP})_2$ exhibits partial solubility in ultra-dried ($<10\text{ ppm H}_2\text{O}$) ethereal solvents like DME and THF, which explains the appearance of these peaks in the spectra of the solvent phase. We have added a description in Supplementary Fig. 18 to clarify that these peaks originate from dissolved $\text{Mg}(\text{DMP})_2$ species rather than from solvent decomposition or new coordination complex.

Supplementary Fig. 18. FT-IR spectra of the (A) THF and (B) G1 solutions before and after immersion of a piece of TMP-treated Mg disc for 10 min. Each solvent was prepared in an ultra-dry ($< 10\text{ ppm H}_2\text{O}$) and moist ($\sim 6500\text{ ppm H}_2\text{O}$) conditions. The absorption peak at $\sim 500\text{ cm}^{-1}$ could be assigned to asymmetric stretching mode of C-Mg-C bond of Me_2Mg as reported in the literature.¹ Therefore, after immersion in the solvent, ~~Me₂Mg~~ released from the TMP-treated Mg is expected to remain dissolved in the dried solvent, whereas it is unlikely to survive in a highly moist solvent.

Fig. R1 FT-IR spectra of (i) the ultra-dried solvent (<10 ppm H₂O) exposed to TMP-treated Mg, and (ii) TMP-treated Mg electrode exposed to moist solvent (~8000 ppm H₂O). The solvent for (A) is G1, and that for (B) is THF. The red spectra, (i), represent the absorbance of dried solution after immersion of TMP-treated Mg. The blue spectra, (ii), correspond to the TMP-treated Mg exposed to moist solution. Synthesized Mg(DMP)₂, (iii), and TMP-treated Mg electrode, (iv), samples are shown as references. The volume of all solutions was 1 mL, and the analysis was performed after a 15 min reaction time.

⇒ In contrast, when water was present (in moist electrolyte), Mg(DMP)₂ becomes insoluble. As shown in Fig. R2, the TMP-treated Mg electrodes immersed in moist G1 or THF display distinct Mg(DMP)₂ peaks, whereas those immersed in ultra-dried G1 or THF show much weaker features. This indicates that once Mg(DMP)₂ adsorbs water molecules from the moist electrolytes, it precipitates onto the electrode the surface, reducing its concentration in the bulk solution. Such behavior is consistent with the hygroscopic nature of Mg(DMP)₂, and its decreased solubility in ethereal solution upon exposure to moisture.

Fig. R2 FT-IR spectra of TMP-treated Mg electrodes after immersion in with (A) G1 and (B) THF. Each solution was prepared in an ultra-dry (< 10 ppm H_2O) and moist (~ 8000 ppm H_2O) conditions. The blue, (i), and the red, (ii), spectra represent the absorbance of TMP-treated Mg electrodes after immersion in moist and dried solvent, respectively. Synthesized $\text{Mg}(\text{DMP})_2$ (iii) and TMP-treated Mg samples (iv) are shown as references. The volume of all solutions was 1 mL, and the analysis was performed after a 15 min reaction time.

⇒ Following the reviewer's suggestion, we added 1 mL of moist electrolyte to 1 mL of ultra-dried solvent that had been previously exposed to the treated Mg and examined the resulting IR spectra (Fig. R3). The peaks associated with $\text{Mg}(\text{DMP})_2$ disappeared completely, and the spectra reverted to those of the pure solvents (THF, G1) or to those observed when the TMP-treated Mg was directly immersed in the moist solvents. This demonstrates that $\text{Mg}(\text{DMP})_2$ species dissolved in the dry solvents react with the added water, becoming insoluble and precipitating from the solution, as also observed on the electrode surface.

Fig. R3 FT-IR spectra of (A) G1 and (B) THF after immersing/removing TMP-treated Mg electrodes. The purple, (i), spectra correspond to the case 1 mL of moist electrolyte (~8000 ppm H₂O) was added to 1 mL of ultra-dried solvent (<10 ppm) that had been previously exposed to treated Mg. The blue, (ii), and red, (iii), spectra represent the absorbance of moist and dried solvent after immersion of TMP-treated Mg, respectively. Synthesized Mg(DMP)₂, (iv), and TMP-treated Mg samples, (v), are shown as references. The volume for (ii), and (iii) was 1 mL, and the analysis was performed after a 15 min reaction time.

- ⇒ Taken together, these results indicate that FTIR spectral changes originate from the presence of dissolved Mg(DMP)₂ in the dry solvents. The experiment further supports that the chemical changes in the solvent arise from the dissolution-precipitation behavior of Mg(DMP)₂, rather than from solvent decomposition or formation of new coordinated species.
- ⇒ We would like to again express our sincere appreciation to all the reviewers for their valuable and insightful comments. We hope that we have adequately addressed all the concerns raised. We believe that our manuscript has significantly improved thanks to the reviewers' **constructive and thoughtful** comments.

Response to Reviewer #4's comments:

The authors have appropriately addressed my concerns, and I recommend that the manuscript be published after one minor revision. I strongly suggest that the authors include the additional FTIR data provided as part of their response to my comments in the revised SI, as it provides important additional information about the origin and evolution of the DMP content in the electrolyte as a function of water content.

⇒ We sincerely appreciate the reviewer's constructive feedback. As suggested, the additional FT-IR data have been included as Supplementary Figures 19, 20, and 21 in the revised Supplementary Information. Detailed explanations of the significance of the additional FT-IR data have been also added to the corresponding figure legends and legend of Supplementary Figure 18.

Supplementary Fig. 18. Fourier transform–Infrared (FT-IR) spectra of the (A) THF and (B) G1 solutions before and after immersion of a piece of trimethyl phosphite (TMP)-treated Mg disc for 10 min. Each solvent was prepared in a highly dried (< 10 ppm H₂O) and moist (~6500 ppm H₂O) conditions. The absorption peak at ~500 cm⁻¹ could be assigned to asymmetric stretching mode of C-Mg-C bond of Me₂Mg as reported.¹ Therefore, after

P
A
G
E

immersion in the solvent, Me_2Mg released from the TMP-treated Mg is expected to remain dissolved in the dried solvent, whereas it is unlikely to survive in a highly moist solvent. Most of the peaks beyond 500 cm^{-1} in the FT-IR spectra of dry solvents exposed to the treated Mg electrodes correspond to the characteristic bands of $\text{Mg}(\text{DMP})_2$, as confirmed in Supplementary Fig. 19. $\text{Mg}(\text{DMP})_2$ exhibits partial solubility in highly dried ($<10\text{ ppm H}_2\text{O}$) DME and THF, which explains the appearance of these peaks in the spectra of the solvent phase. In contrast, when water is present (in moist electrolyte), $\text{Mg}(\text{DMP})_2$ becomes insoluble. As shown in Supplementary Fig. 20, the TMP-treated Mg electrodes immersed in moist G1 or THF display distinct $\text{Mg}(\text{DMP})_2$ peaks, whereas those immersed in highly dried G1 or THF show much weaker features. This indicates that once $\text{Mg}(\text{DMP})_2$ adsorbs water molecules from the moist electrolytes, it precipitates onto the electrode the surface, reducing its concentration in the bulk solution. Such behavior is consistent with the hygroscopic nature of $\text{Mg}(\text{DMP})_2$, and its decreased solubility in ethereal solutions upon exposure to moisture. To confirm this, 1 mL of moist electrolyte was added to 1 mL of highly dried solvent that had been previously exposed to the treated Mg and examined the resulting FT-IR spectra (Supplementary Fig. 21). The peaks associated with $\text{Mg}(\text{DMP})_2$ disappeared completely, and the spectra reverted to those of the pure solvents or to those observed when the TMP-treated Mg was directly immersed in the moist solvents. This demonstrates that $\text{Mg}(\text{DMP})_2$ species dissolved in the dry solvents react with the added water, becoming insoluble and precipitating from the solution, as also observed on the electrode surface.

Supplementary Fig. 19. Fourier transform–infrared (FT-IR) spectra of (i) the highly dried solvent (<10 ppm H₂O) exposed to trimethyl phosphite (TMP)-treated Mg, and (ii) TMP-treated Mg electrode exposed to moist solvent (~8000 ppm H₂O). The solvent for (A) is G1, and that for (B) is THF. The red spectra, (i), represent the absorbance of dried solution after immersion of TMP-treated Mg. The blue spectra, (ii), correspond to the TMP-treated Mg exposed to moist solution. Synthesized Mg(DMP)₂, (iii), and TMP-treated Mg electrode, (iv), samples are shown as references. The volume of all solutions was 1 mL, and the analysis was performed after a 15 min reaction time.

Supplementary Fig. 20. Fourier transform–infrared (FT-IR) spectra of trimethyl phosphate (TMP)-treated Mg electrodes after immersion in with (A) G1 and (B) THF. Each solution was prepared in a highly dried (< 10 ppm H₂O) and moist (~8000 ppm H₂O) conditions. The blue, (i), and the red, (ii), spectra represent the absorbance of TMP-treated Mg electrodes after immersion in moist and dried solvent, respectively. Synthesized Mg(DMP)₂ (iii) and TMP-treated Mg samples (iv) are shown as references. The volume of all solutions was 1 mL, and the analysis was performed after a 15 min reaction time.

Supplementary Fig. 21. Fourier transform-infrared (FT-IR) spectra of (A) G1 and (B) THF after immersing / removing trimethyl phosphate (TMP)-treated Mg electrodes. The purple, (i), spectra correspond to the case 1 mL of moist electrolyte (~8000 ppm H₂O) was added to 1 mL of highly dried solvent (<10 ppm) that had been previously exposed to treated Mg. The blue, (ii), and red, (iii), spectra represent the absorbance of moist and dried solvent after immersion of TMP-treated Mg, respectively. Synthesized Mg(DMP)₂, (iv), and TMP-treated Mg samples, (v), are shown as references. The volume for (ii), and (iii) was 1 mL, and the analysis was performed after a 15 min reaction time.

The authors research important point, namely, the improvement of the compatibility of Mg metal anodes by surface treatment with alkyl phosphate additives. They showcase that use of TMP surface treatment can improve Mg metal anode compatibility with dry room atmosphere and wet electrolytes. The topic is timely and addresses an important research question. However, the same additives have already been reported in field of Mg batteries, with the main mechanism presumably being the altered solvation shell of Mg^{2+} ions.^{1,2} I think the authors should aim and clarify difference between their and literature reports to better assign contributions to either bulk electrolyte solvation or Mg metal interphase. I think this should significantly strengthen the impact of the work. I have also following more specific comments.

- 1) I would disagree that moisture sensitivity of Mg metal anode is the key constraint preventing Mg battery market entry. I believe that many issues like developing suitable cathode materials improving long-term cycling performance and demonstrating high-energy density on practical cell level need to be addressed before. Hence, I would suggest rephrasing the abstract and introduction.
- 2) The authors confirm formation of $Mg(DMP)_2$ on top of the Mg metal anode, which acts as scavenger during electrochemical cycling. It would be important to measure the water content in the wet electrolyte after cycling to evaluate scavenging effect during cycling. Did they also test several different electrolyte amounts in their cells.
- 3) The authors should showcase the effect of TMP also on better performing weakly coordinating electrolytes that have shown much better Mg plating/stripping electrochemical performance.³⁻⁵ Among them, MgAlhfp has already showcased very high water tolerance.⁵
- 4) More focus should be on the cycling in chloride-free electrolytes like E3 since chloride-free electrolytes are the only realistic option for practical Mg batteries. Most of their cycling result are shown on the E1 electrolyte, which contains $MgCl_2$ and it was already shown that $MgCl_2$ - $Mg(TFSI)_2$ has better water tolerance than plain $Mg(TFSI)_2$.⁶
- 5) There is a lack of Mg metal interphase characterization during and after electrochemical cycling. I believe that surface characterization through electron microscopy, XPS, IR.... It could add important insight into the changes in the interphase during cycling.

References:

- (1) Zhao, W.; Pan, Z.; Zhang, Y.; Liu, Y.; Dou, H.; Shi, Y.; Zuo, Z.; Zhang, B.; Chen, J.; Zhao, X.; Yang, X. Tailoring Coordination in Conventional Ether-Based Electrolytes for Reversible Magnesium-Metal Anodes. *Angewandte Chemie* **2022**, *61* (30), e202205187. <https://doi.org/10.1002/ange.202205187>.

- (2) Li, C.; Guha, R. D.; Shyamsunder, A.; Persson, K. A.; Nazar, L. F. A Weakly Ion Pairing Electrolyte Designed for High Voltage Magnesium Batteries. *Energy Environ Sci* **2024**, *17* (1), 190–201. <https://doi.org/10.1039/D3EE02861E>.
- (3) Mandai, T.; Youn, Y.; Tateyama, Y. Remarkable Electrochemical and Ion-Transport Characteristics of Magnesium-Fluorinated Alkoxyaluminate-Diglyme Electrolytes for Magnesium Batteries. *Mater Adv* **2021**, *2* (19), 6283–6296. <https://doi.org/10.1039/d1ma00448d>.
- (4) Zhao-Karger, Z.; Gil Bardaji, M. E.; Fuhr, O.; Fichtner, M. A New Class of Non-Corrosive, Highly Efficient Electrolytes for Rechargeable Magnesium Batteries. *J. Mater. Chem. A* **2017**, *5*, 10815–10820. <https://doi.org/10.1039/C7TA02237A>.
- (5) Pavčnik, T.; Lozinšek, M.; Pirnat, K.; Vizintin, A.; Mandai, T.; Aurbach, D.; Dominko, R.; Bitenc, J. On the Practical Applications of the Magnesium Fluorinated Alkoxyaluminate Electrolyte in Mg Battery Cells. *ACS Appl Mater Interfaces* **2022**, *14* (23), 26766–26774. <https://doi.org/10.1021/acsami.2c05141>.
- (6) Connell, J. G.; Genorio, B.; Lopes, P. P.; Strmcnik, D.; Stamenkovic, V. R.; Markovic, N. M. Tuning the Reversibility of Mg Anodes via Controlled Surface Passivation by H₂O/Cl – in Organic Electrolytes. *Chemistry of Materials* **2016**, *28* (22), 8268–8277. <https://doi.org/10.1021/acs.chemmater.6b03227>.